# Near-Optimal Distributed Minimax Optimization under the Second-Order Similarity

**Qihao Zhou**
School of Data Science, Fudan University
`zhouqh20@fudan.edu.cn`

**Haishan Ye**
School of Management, Xi'an Jiaotong University
SGIT AI Lab, State Grid Corporation of China
`yehaishan@xjtu.edu.cn`

**Luo Luo**[*]
School of Data Science, Fudan University
Shanghai Key Laboratory for Contemporary Applied Mathematics
`luoluo@fudan.edu.cn`

## Abstract

This paper considers the distributed convex-concave minimax optimization under the second-order similarity. We propose stochastic variance-reduced optimistic gradient sliding (SVOGS) method, which takes the advantage of the finite-sum structure in the objective by involving mini-batch client sampling and variance reduction. We prove SVOGS can achieve the $\varepsilon$-duality gap within communication rounds of $\mathcal{O}(\delta D^2/\varepsilon)$, communication complexity of $\tilde{\mathcal{O}}(n + \sqrt{n}\delta D^2/\varepsilon)$, and local gradient calls of $\tilde{\mathcal{O}}(n + (\sqrt{n}\delta + L)D^2/\varepsilon \log(1/\varepsilon))$, where $n$ is the number of nodes, $\delta$ is the degree of the second-order similarity, $L$ is the smoothness parameter, and $D$ is the diameter of the constraint set. We can verify that all of above complexity (nearly) matches the corresponding lower bounds. For the specific $\mu$-strongly-convex-$\mu$-strongly-convex case, our algorithm has the upper bounds on communication rounds, communication complexity, and local gradient calls of $\mathcal{O}(\delta/\mu \log(1/\varepsilon))$, $\mathcal{O}((n + \sqrt{n}\delta/\mu) \log(1/\varepsilon))$, and $\tilde{\mathcal{O}}(n + (\sqrt{n}\delta + L)/\mu) \log(1/\varepsilon))$ respectively, which are also nearly tight. Furthermore, we conduct the numerical experiments to show the empirical advantages of the proposed method.

## 1 Introduction

We study the distributed minimax optimization problem

$$\min_{x \in \mathcal{X}} \max_{y \in \mathcal{Y}} f(x, y) := \frac{1}{n} \sum_{i=1}^{n} f_i(x, y), \tag{1}$$

where $f_i$ is the differentiable local function associated with the $i$-th node, and $\mathcal{X} \subseteq \mathbb{R}^{d_x}$ and $\mathcal{Y} \subseteq \mathbb{R}^{d_y}$ are the constraint sets. We are interested in the centralized setting, where there are one server node and $n - 1$ client nodes that collaboratively solve the minimax problem. Without loss of generality, we assume the function $f_1$ is located on the server node and the functions $f_2, \ldots, f_n$ are located on the client nodes. This formulation is a cornerstone in the study of game theory, aiming to achieve the Nash equilibrium [12, 44]. It covers a lot of applications such as signal processing [23], optimal control [41], adversarial learning [44], robust regression [15, 35] and portfolio management [52].

---

[*]The corresponding author

38th Conference on Neural Information Processing Systems (NeurIPS 2024).

We focus on the first-order optimization methods for solving convex-concave minimax problem. The classical full-batch approaches including extra-gradient (EG) method [24], forward-backward-forward (FBF) [51], optimistic gradient descent ascent (OGDA) [43], dual extrapolation [39] and so forth [33, 34, 38] achieve the optimal first-order oracle complexity under the assumption of Lipschitz continuous gradient [17, 42, 55]. For the objective with finite-sum structure, the stochastic variance reduced methods [1, 10, 14, 30, 53] can reduce the cost of per iteration by using the inexact gradient and lead to the better overall computational cost than full-batch methods. It is natural to design the parallel iteration schemes by directly using above ideas to reduce the computational time in distributed setting.

The communication complexity is a primary bottleneck in distributed optimization. The local functions in machine learning applications typically exhibit homogeneity [3, 15, 18], which is helpful to improve the communication efficiency. One common measure used to describe relationships among local functions is the second-order similarity, e.g., the Hessian of each local function differs by a finite quantity from the Hessian of global objective. Based on such characterization, several communication efficient distributed optimization methods have been established [4, 5, 8, 19, 20, 25, 29, 46, 48, 50, 56]. The highlight of these methods is their communication complexity bounds mainly depend on the degree of second-order similarity, which is potentially much tighter than the results depend on the smoothness parameter [6, 7, 11, 13, 16, 21, 22, 25, 26, 28, 32, 36, 37, 47].

Recently, Khaled and Jin [20], Lin et al. [29] showed iterations with partial participation can further reduce the communication complexity, improving the dependence on the number of nodes. They proposed stochastic variance reduced proximal point methods for convex optimization, which allow only one of clients to participate into the communication in the most of rounds. Additionally, Beznosikov et al. [8] combined partial participation with forward-backward-forward based method, reducing volume of communication complexity for minimax optimization. However, these methods [8, 20, 29] increase the communication rounds, which result in more expensive time cost in communication than the full participation strategies [5, 25]. In other words, the partial participation methods [8, 20, 29] only reduce the overall volume of information exchanged among the nodes, while the advantage of parallel communication enjoyed in full participation methods is damaged.

In this paper, we propose a novel distributed minimax optimization method, called stochastic variance-reduced optimistic gradient sliding (SVOGS), which uses the mini-batch client sampling to balance communication rounds, communication complexity, and computational complexity. We prove SVOGS simultaneously achieves the (near) optimal communication complexity, communication rounds, and local gradient calls for convex-concave minimax problem under the assumption of second-order similarity. We also conduct numerical experiments to show the superiority of SVOGS.

## 2 Preliminaries

We focus on the distributed optimization in client-sever framework for solving minimax problem (1). The notation $f_i$ presents the local function on the $i$-th node. We assume the function $f_1$ is located on the server and the other individuals are located on clients. We stack variables $x \in \mathbb{R}^{d_x}$ and $y \in \mathbb{R}^{d_y}$ as the vector $z = [x; y] \in \mathbb{R}^d$, where $d = d_x + d_y$. We let $\mathcal{Z} := \mathcal{X} \times \mathcal{Y} \subseteq \mathbb{R}^d$ and define the projection operator $\mathcal{P}_{\mathcal{Z}}(v) := \arg\min_{z \in \mathcal{Z}} \|z - v\|$ for given $v \in \mathbb{R}^d$. We also denote the vector functions $F_i : \mathbb{R}^d \to \mathbb{R}^d$ and $F : \mathbb{R}^d \to \mathbb{R}^d$ as

$$F_i(z) := \begin{bmatrix} \nabla_x f_i(x, y) \\ -\nabla_y f_i(x, y) \end{bmatrix} \qquad \text{and} \qquad F(z) := \frac{1}{n} \sum_{i=1}^n F_i(z).$$

We consider the following common assumptions for our minimax problem.

**Assumption 1.** *We suppose the constraint set $\mathcal{Z} \subseteq \mathbb{R}^d$ is a non-empty, closed, and convex.*

**Assumption 2.** *We suppose the constraint set $\mathcal{Z} \subseteq \mathbb{R}^d$ is bounded by diameter $D > 0$, i.e., we have $\|z_1 - z_2\| \leq D$ for all $z_1, z_2 \in \mathcal{Z}$.*

**Assumption 3.** *We suppose each local function $f_i : \mathbb{R}^{d_x} \times \mathbb{R}^{d_y} \to \mathbb{R}$ is smooth, i.e., there exists $L > 0$ such that $\|F_i(z_1) - F_i(z_2)\| \leq L \|z_1 - z_2\|$ for all $i \in [n]$ and $z_1, z_2 \in \mathbb{R}^d$ .*

**Assumption 4.** *We suppose each differentiable local function $f_i : \mathbb{R}^{d_x} \times \mathbb{R}^{d_y} \to \mathbb{R}$ is convex-concave, i.e., we have $f_i(x, y) \geq f_i(x', y) + \langle \nabla_x f_i(x', y), x - x' \rangle$ and $f_i(y) \leq f_i(y') + \langle \nabla_y f_i(y'), y - y' \rangle$ for all $i \in [n]$, $x, x' \in \mathbb{R}^{d_x}$ and $y, y' \in \mathbb{R}^{d_y}$.*

**Assumption 5.** *We suppose the global objective $f : \mathbb{R}^{d_x} \times \mathbb{R}^{d_y} \to \mathbb{R}$ is strongly-convex-strongly-concave, i.e., there exists $\mu > 0$ such that the function $f(x,y) - \frac{\mu}{2}\|x\|^2 + \frac{\mu}{2}\|y\|^2$ is convex-concave.*

Besides, we introduce the assumption of second-order similarity to measure the homogeneity in local functions [5, 8, 19, 20].

**Assumption 6.** *The local functions $f_1, \ldots, f_n : \mathbb{R}^{d_x} \times \mathbb{R}^{d_y} \to \mathbb{R}$ are twice differentiable and hold the $\delta$-second-order similarity, i.e., there exists $\delta > 0$ such that*

$$\left\| \nabla^2 f_i(x,y) - \nabla^2 f(x,y) \right\| \leq \delta$$

*for all $i \in [n]$, $x \in \mathbb{R}^{d_x}$ and $y \in \mathbb{R}^{d_y}$.*

We measure the sub-optimality of the approximate solution $z = (x,y) \in \mathcal{Z}$ by duality gap, that is

$$\mathrm{Gap}(x,y) := \max_{y' \in \mathcal{Y}} f(x,y') - \min_{x' \in \mathcal{X}} f(x',y).$$

We also consider the criterion of the gradient mapping for given $z = (x,y) \in \mathcal{Z}$ [31, 40, 54], that is,

$$\mathscr{F}_\tau(z) = \frac{z - \mathcal{P}_\mathcal{Z}(z - \tau F(z))}{\tau},$$

where $\tau > 0$. The gradient mapping $\mathscr{F}_\tau(z)$ is a natural extension of gradient operator $F(z)$. Noticing that we have $\mathscr{F}_\tau(z) = F(z)$ if the problem is unconstrained (i.e., $\mathcal{Z} = \mathbb{R}^d$), and the condition $\mathscr{F}_\tau(z) = 0$ is equivalent to the point $z$ is a solution of the problem. Compared with the duality gap, the norm of gradient mapping is a more popular measure in empirical studied since it is easy to achieve in practice.

For the specific strongly-convex-strongly-concave case, we can also measure the sub-optimality by the square of Euclidean distance to the unique solution $z^* = (x,y) \in \mathcal{Z}$, that is

$$\|z - z^*\|^2 = \|x - x^*\|^2 + \|y - y^*\|^2.$$

Moreover, we use notations $\mathcal{O}(\cdot)$, $\Theta(\cdot)$ and $\Omega(\cdot)$ to hide constants which do not depend on parameters of the problem, and notations $\tilde{\mathcal{O}}(\cdot)$, $\tilde{\Theta}(\cdot)$, and $\tilde{\Omega}(\cdot)$ to additionally hide the logarithmic factors of $n$, $L$, $\mu$ and $\delta$.

## 3 Related Work

For convex-concave minimax optimization, the full batch first-order methods [24, 33, 34, 38, 39, 43, 51] can achieve $\varepsilon$-duality gap within at most $\mathcal{O}(LD^2/\varepsilon)$ iterations. Applying these idea to distributed setting naturally leads to the communication rounds of $\mathcal{O}(LD^2/\varepsilon)$ and each round requires all of the $n$ nodes to compute and communicate their local gradient.

In a seminar work, Beznosikov et al. [5] proposed Star Min-Max Data Similarity (SMMDS) algorithm, which additionally consider the second-orders similarity (Assumption 6) by involving gradient sliding technique [27, 45]. The SMMDS requires communication rounds of $\mathcal{O}(\delta D^2/\varepsilon)$, which benefits from the homogeneity in local functions. Each round of this method needs to communicate/compute the local gradient of all $n$ nodes, and perform the local updates on the server within $\tilde{\mathcal{O}}(L/\delta \log(1/\varepsilon))$ local iterations, which results in the overall communication complexity of $\mathcal{O}(n\delta D^2/\varepsilon)$ and local gradient complexity of $\tilde{\mathcal{O}}((n\delta + L)D^2/\varepsilon \log(1/\varepsilon))$. Later, Kovalev et al. [25] introduced extra-gradient sliding (EGS), which further improves the local gradient complexity to $\mathcal{O}((n\delta + L)D^2/\varepsilon)$. It is worth pointing out that the communication rounds of $\mathcal{O}(\delta D^2/\varepsilon)$ achieved by SMMDS and EGS matches the lower complexity bound under the second-order similarity assumption [5]. However, these methods enforce all nodes to participate into communication in every round, which does not sufficiently take the advantage of finite-sum structure in the objective.

Recently, Beznosikov et al. [8] proposed Three Pillars Algorithm with Partial Participation (TPAPP), which uses the variance-reduced forward-backward-forward method [1, 10] to encourage only one of clients participate into the communication in most of the rounds. The TPAPP can achieve point $z \in \mathbb{R}^d$ such that $\mathbb{E}[\|F(z)\|^2] \leq \varepsilon$ for unconstrained case within the communication rounds of $\mathcal{O}(n\delta^2 D^2/\varepsilon)$,

Table 1: The complexity of achieving $\mathbb{E}[\mathrm{Gap}(x,y)] \leq \varepsilon$ in convex-concave case.

| Methods | Communication Rounds | Communication Complexity | Local Gradient Complexity |
|---|---|---|---|
| EG [24] | $\mathcal{O}\big(\frac{LD^2}{\varepsilon}\big)$ | $\mathcal{O}\big(\frac{nLD^2}{\varepsilon}\big)$ | $\mathcal{O}\big(\frac{nLD^2}{\varepsilon}\big)$ |
| SMMDS [5] | $\mathcal{O}\big(\frac{\delta D^2}{\varepsilon}\big)$ | $\mathcal{O}\big(\frac{n\delta D^2}{\varepsilon}\big)$ | $\tilde{\mathcal{O}}\big(\frac{(n\delta+L)D^2}{\varepsilon}\log\frac{1}{\varepsilon}\big)$ |
| EGS [25] | $\mathcal{O}\big(\frac{\delta D^2}{\varepsilon}\big)$ | $\mathcal{O}\big(\frac{n\delta D^2}{\varepsilon}\big)$ | $\mathcal{O}\big(\frac{(n\delta+L)D^2}{\varepsilon}\big)$ |
| SVOGS (Algorithm 1) | $\mathcal{O}\big(\frac{\delta D^2}{\varepsilon}\big)$ | $\mathcal{O}\big(n+\frac{\sqrt{n}\delta D^2}{\varepsilon}\big)$ | $\tilde{\mathcal{O}}\big(n+\frac{(\sqrt{n}\delta+L)D^2}{\varepsilon}\log\frac{1}{\varepsilon}\big)$ |
| Lower Bounds (Theorem 3,4,5) | $\Omega\big(\frac{\delta D^2}{\varepsilon}\big)$ | $\Omega\big(n+\frac{\sqrt{n}\delta D^2}{\varepsilon}\big)$ | $\Omega\big(n+\frac{(\sqrt{n}\delta+L)D^2}{\varepsilon}\big)$ |

Table 2: The complexity of achieving $\mathbb{E}[\|x-x^*\|^2 + \|y-y^*\|^2] \leq \varepsilon$ in the strongly-convex-strongly-concave case. [†]These methods use permutation compressors [49], which require the assumption of $d > n$. [♯]The complexity of TPAPP depends on local iterations number $H$, where "TPAPP (a)" and "TPAPP (b)" correspond to $H=\lceil L/(\sqrt{n}\delta)\rceil$ and $H=\lceil 8\log(40nL/\mu)\rceil$ respectively.

| Methods | Communication Rounds | Communication Complexity | Local Gradient Complexity |
|---|---|---|---|
| EG [24] | $\mathcal{O}\big(\frac{L}{\mu}\log\frac{1}{\varepsilon}\big)$ | $\mathcal{O}\big(\frac{nL}{\mu}\log\frac{1}{\varepsilon}\big)$ | $\mathcal{O}\big(\frac{nL}{\mu}\log\frac{1}{\varepsilon}\big)$ |
| SMMDS [5] | $\mathcal{O}\big(\frac{\delta}{\mu}\log\frac{1}{\varepsilon}\big)$ | $\mathcal{O}\big(\frac{n\delta}{\mu}\log\frac{1}{\varepsilon}\big)$ | $\tilde{\mathcal{O}}\big(\frac{n\delta+L}{\mu}\log\frac{1}{\varepsilon}\big)$ |
| EGS [25] | $\mathcal{O}\big(\frac{\delta}{\mu}\log\frac{1}{\varepsilon}\big)$ | $\mathcal{O}\big(\frac{n\delta}{\mu}\log\frac{1}{\varepsilon}\big)$ | $\mathcal{O}\big(\frac{n\delta+L}{\mu}\log\frac{1}{\varepsilon}\big)$ |
| OMASHA [4][†] | $\mathcal{O}\big(\frac{L}{\mu}\log\frac{1}{\varepsilon}\big)$ | $\mathcal{O}\big((n+\frac{\sqrt{n}\delta+L}{\mu})\log\frac{1}{\varepsilon}\big)$ | $\mathcal{O}\big(\frac{nL}{\mu}\log\frac{1}{\varepsilon}\big)$ |
| TPA [8][†] | $\mathcal{O}\big((n+\frac{\sqrt{n}\delta}{\mu})\log\frac{1}{\varepsilon}\big)$ | $\mathcal{O}\big((n+\frac{\sqrt{n}\delta}{\mu})\log\frac{1}{\varepsilon}\big)$ | $\mathcal{O}\big((n+\frac{\sqrt{n}L}{\delta}+\frac{L}{\mu})\log\frac{1}{\varepsilon}\big)$ |
| TPAPP (a) [8][♯] | $\mathcal{O}\big((n+\frac{\sqrt{n}\delta}{\mu})\log\frac{1}{\varepsilon}\big)$ | $\mathcal{O}\big((n+\frac{\sqrt{n}\delta}{\mu})\log\frac{1}{\varepsilon}\big)$ | $\mathcal{O}\big((n+\frac{\sqrt{n}L}{\delta}+\frac{L}{\mu})\log\frac{1}{\varepsilon}\big)$ |
| TPAPP (b) [8][♯] | $\mathcal{O}\big((n+\frac{\sqrt{n}\delta+L}{\mu})\log\frac{1}{\varepsilon}\big)$ | $\mathcal{O}\big((n+\frac{\sqrt{n}\delta+L}{\mu})\log\frac{1}{\varepsilon}\big)$ | $\tilde{\mathcal{O}}\big((n+\frac{\sqrt{n}\delta+L}{\mu})\log\frac{1}{\varepsilon}\big)$ |
| SVOGS (Algorithm 1) | $\mathcal{O}\big(\frac{\delta}{\mu}\log\frac{1}{\varepsilon}\big)$ | $\mathcal{O}\big((n+\frac{\sqrt{n}\delta}{\mu})\log\frac{1}{\varepsilon}\big)$ | $\tilde{\mathcal{O}}\big((n+\frac{\sqrt{n}\delta+L}{\mu})\log\frac{1}{\varepsilon}\big)$ |
| Lower Bounds ([5, 8], Theorem 7) | $\Omega\big(\frac{\delta}{\mu}\log\frac{1}{\varepsilon}\big)$ | $\Omega\big((n+\frac{\sqrt{n}\delta}{\mu})\log\frac{1}{\varepsilon}\big)$ | $\Omega\big((n+\frac{\sqrt{n}\delta+L}{\mu})\log\frac{1}{\varepsilon}\big)$ |

communication complexity of $\mathcal{O}(n\delta^2 D^2/\varepsilon)$, and local gradient complexity of $\mathcal{O}(n^2\delta^4 L^2 D^6 \varepsilon^{-3})$.[2] The theoretical analysis of TPAPP for the constrained problem requires the objective being strongly-convex-strongly-concave. In addition, we can also reduce the communication complexity by using the permutation compressors [49] for high-dimensional problem [4, 8], which achieves the similar complexity to existing partial participation methods [8].

We present the complexity of existing methods and compare them with our results in both general convex-concave case and strongly-convex-strongly-concave case in Table 1-3.

## 4 Stochastic Variance-Reduced Optimistic Gradient Sliding

We propose stochastic variance-reduced optimistic gradient sliding (SVOGS) method in Algorithm 1.

The design of our algorithm starts from reformulating problem (1) as follows

$$\min_{x\in\mathcal{X}}\max_{y\in\mathcal{Y}} f(x,y) := \underbrace{\frac{1}{n}\sum_{i=1}^{n}(f_i(x,y)-f_1(x,y))}_{g(x,y):=f(x,y)-f_1(x,y)} + f_1(x,y). \tag{3}$$

The idea of gradient sliding [27] on minimax optimization can be viewed as iteratively solving the surrogate of problem (3) within the quadratic approximation of $g(x,y)$ [5, 8, 25]. Recall that the

---

[2]Although the complexity of TPAPP for achieving $\mathbb{E}[\|F(z)\|]^2 \leq \varepsilon$ is established for the unconstrained case, its analysis additionally assume that the sequences generated by the algorithm are bounded by $D > 0$ [8, Theorem 5.12].

Table 3: The complexity of achieving $\mathbb{E}[\|\mathscr{F}_\tau(x,y)\|^2] \leq \varepsilon$ in convex-concave case. §The TPAPP additionally assumes $\mathcal{Z} = \mathbb{R}^d$ and the sequence generated by the algorithm is bounded by $D > 0$.

| Methods | Communication Rounds | Communication Complexity | Local Gradient Complexity |
|---|---|---|---|
| TPAPP [8]§ | $\mathcal{O}\big(\frac{n\delta^2 D^2}{\varepsilon}\big)$ | $\mathcal{O}\big(\frac{n\delta^2 D^2}{\varepsilon}\big)$ | $\mathcal{O}\big(\frac{n^2\delta^4 L^2 D^6}{\varepsilon^3}\big)$ |
| SVOGS (Algorithm 1) | $\tilde{\mathcal{O}}\big(\frac{\delta D}{\sqrt{\varepsilon}} \log \frac{1}{\varepsilon}\big)$ | $\tilde{\mathcal{O}}\big((n + \frac{\sqrt{n}\delta D}{\sqrt{\varepsilon}}) \log \frac{1}{\varepsilon}\big)$ | $\tilde{\mathcal{O}}\big((n + \frac{(\sqrt{n}\delta+L)D}{\sqrt{\varepsilon}}) \log \frac{1}{\varepsilon}\big)$ |

---

**Algorithm 1** Stochastic Variance-Reduced Optimistic Gradient Sliding (SVOGS)

1: **Input:** initial point $z^0 = (x^0, y^0) \in \mathcal{Z}$, step size $\eta$, accuracy $\{\varepsilon_k\}_{k=1}^K$, communication rounds $K$, mini-batch size $b$, probability $p \in (0, 1]$, weights $\alpha, \gamma \in (0, 1)$;

2: **Initialization:** $w^{-1} = z^{-1} = w^0 = z^0 = (x^0, y^0) \in \mathcal{Z}$, $z_i^0 = z^0$ for all $i \in [n]$;

3: **for** $k = 0, 1, 2, \ldots, K - 1$ **do**

4:   $\bar{z}^k = (1 - \gamma)z^k + \gamma w^k$;

5:   Sample $\mathcal{S}^k = \{j_1^k, \ldots, j_b^k\}$ uniformly and independently from $[n]$;

6:   $\delta^k = F(w^{k-1}) - F_1(w^{k-1}) + \frac{1}{b} \sum_{j \in \mathcal{S}^k} \big(F_j(z^k) - F_1(z^k) - F_j(w^{k-1}) + F_1(w^{k-1})\big)$

$\quad + \frac{\alpha}{b} \sum_{j \in \mathcal{S}^k} \big(F_j(z^k) - F_1(z^k) - F_j(z^{k-1}) + F_1(z^{k-1})\big)$;

7:   $v^k = \bar{z}^k - \eta\delta^k$;

8:   Find $u^k \in \mathbb{R}^d$ such that $\|u^k - \hat{u}^k\|^2 \leq \varepsilon_k$, where $\hat{u}^k$ is the solution of the problem

$$\min_{\hat{x} \in \mathcal{X}} \max_{\hat{y} \in \mathcal{Y}} \left\{ f_1(\hat{x}, \hat{y}) + \frac{1}{2\eta}\|\hat{x} - v_x^k\|^2 - \frac{1}{2\eta}\|\hat{y} - v_y^k\|^2 \right\}; \tag{2}$$

9:   $z^{k+1} = u^k$;

10:   $w^{k+1} = \begin{cases} z^{k+1} & \text{with probability } p, \\ w^k & \text{with probability } 1 - p. \end{cases}$

11: **end for**

---

optimistic gradient descent ascent (OGDA) method [34, 43] iterates with

$$z^{k+1} = \mathcal{P}_{\mathcal{Z}}\big(z^k - \eta\underbrace{(F(z^k) + F(z^k) - F(z^{k-1}))}_{\text{optimistic gradient}}\big), \tag{4}$$

where $\eta > 0$ is the step size. It is well-known that OGDA achieves optimal convergence rate under the first-order smoothness assumption [42, 55], which motivated us construct the quadratic approximation of $g(x, y)$ by using the optimistic gradient of $g$ at $(x^k, y^k)$ in the linear terms, that is

$$g(x, y) \approx \hat{g}(x, y)$$
$$= g(x^k, y^k) + \langle \underbrace{\nabla_x g(x^k, y^k) + \nabla_x g(x^k, y^k) - \nabla_x g(x^{k-1}, y^{k-1})}_{\text{optimistic gradient with respect to } x}, x - x^k \rangle + \frac{1}{2\eta}\|x - x^k\|^2$$
$$+ \langle \underbrace{\nabla_y g(x^k, y^k) + \nabla_y g(x^k, y^k) - \nabla_y g(x^{k-1}, y^{k-1})}_{\text{optimistic gradient with respect to } y}, y - y^k \rangle - \frac{1}{2\eta}\|y - y^k\|^2. \tag{5}$$

Applying approximation (5) to formulation (3), we obtain the optimistic gradient sliding (OGS), which iteratively solve the sub-problem

$$(x^{k+1}, y^{k+1}) \approx \arg\min_{\hat{x} \in \mathcal{X}} \max_{\hat{y} \in \mathcal{Y}} \hat{g}(\hat{x}, \hat{y}) + f_1(\hat{x}, \hat{y}). \tag{6}$$

We can verify function $g(x, y)$ is $\delta$-smooth under Assumption 6, which indicates taking $\eta = \Theta(1/\delta)$ and solving the sub-problem sufficiently accurate can find an $\varepsilon$-suboptimal solution within the iteration numbers of $\mathcal{O}(\delta D^2/\varepsilon)$ and $\mathcal{O}(\delta/\mu \log(1/\varepsilon))$ for the convex-concave case and the strongly-convex-strongly-concave case respectively (see Section 5). The dependence on $\delta$ implies OGS benefits from the second-order similarity in local functions, while each of its iteration requires the communication and the computation of the exact gradient of $f(x, y)$ within the complexity of $\mathcal{O}(n)$.

The key idea to improve the cost in each iteration is involving the mini-batch client sampling and variance reduction with momentum [2, 26]. Specifically, we estimate the optimistic gradient in formulation (5) as follows

$$G(z^k) + G(z^k) - G(z^{k-1}) \approx \frac{1}{|\mathcal{S}^k|} \sum_{j \in \mathcal{S}^k} \big( G(w^{k-1}) + G_j(z^k) - G_j(w^{k-1}) + \underbrace{\alpha(G_j(z^k) - G_j(z^{k-1}))}_{\text{momentum term}} \big), \tag{7}$$

where $G(z) := F(z) - F_1(z)$, $\mathcal{S}^k \subseteq [n]$ is the random index set, $w^k$ is the snapshot point which is updated infrequently in iterations, and $\alpha \in (0, 1)$ is the momentum parameter. Applying the optimistic gradient estimation (7) to formulations (5)-(6), we achieve our stochastic variance-reduced optimistic gradient sliding (SVOGS) method (Algorithm 1).

The proposed SVOGS enjoys the mini-batch partial participation in the steps communication and computation in most of rounds, which is the main difference between SVOGS and existing methods [5, 8, 26]. Concretely, taking the mini-batch size $|\mathcal{S}^k| = \Theta(\sqrt{n})$ for SVOGS can simultaneously balance communication rounds, communication complexity and local gradient complexity. The SVOGS keeps both the benefit of parallel communication like full participation methods (i.e., SMMDS [5] and EGS [25]) and the low communication cost like existing participation methods (i.e., TPAPP [8]). Additionally, the communication advantage of SVOGS also makes the algorithm achieves better local gradient complexity than state-of-the-arts [5, 8, 26].

## 5 The Complexity Analysis

In this section, we provide the complexity analysis of proposed SVOGS (Algorithm 1) to show its superiority. In particular, we let $\mu = 0$ for the convex-concave case to the ease of presentation.

We analyze the convergence of SVOGS (Algorithm 1) by establishing the Lyapunov function

$$\Phi^k := \left( \frac{1-\gamma}{\eta} + \frac{\mu}{2} \right) \|z^k - z^*\|^2 + 2\langle F(z^{k-1}) - F_1(z^{k-1}) - F(z^k) + F_1(z^k), z^k - z^* \rangle$$

$$+ \frac{1}{64\eta} \|z^k - z^{k-1}\|^2 + \frac{\gamma}{4\eta} \|w^{k-1} - z^k\|^2 + \frac{(2\gamma + \eta\mu)}{2p\eta} \|w^k - z^*\|^2, \tag{8}$$

where we take weight $\gamma \le 1/8$ and the step size $\eta \le 1/(32\delta)$ which always guarantees $\Phi^k \ge 0$ by using Young's inequality and the similarity assumption (see detailed proof in Appendix B).

We show that the decrease of Lyapunov function in expectation as follows.

**Lemma 1.** *Suppose Assumptions 1, 3, 4, and 6 hold with $0 \le \mu \le \delta \le L$, running SVOGS (Algorithm 1) with $\eta \le \min\{1/\mu, 1/(32\delta)\}$, $\alpha = \max\left\{1 - \eta\mu/(6(1 - \gamma)), 1 - p\eta\mu/(2\gamma + \eta\mu)\right\}$, $\gamma \le 1/8$, $256\eta^2\delta^2\alpha^2(b + 1)/b \le \alpha$, $4\eta\delta^2/b \le \alpha\gamma/(4\eta)$, and $\varepsilon_k \le c^{-1} \min\left\{\|\hat{u}^k - z^k\|, \|\hat{u}^k - z^k\|^2\right\}$ for some $c = \mathrm{poly}(\mu, \delta)$, then we have*

$$\mathbb{E}[\Phi^{k+1}] \le \max\left\{ 1 - \frac{\eta\mu}{6(1 - \gamma)}, 1 - \frac{p\eta\mu}{2\gamma + \eta\mu} \right\} \mathbb{E}[\Phi^k] - \frac{1}{16\eta} \mathbb{E}\left[\|z^k - \hat{u}^k\|^2\right] - \frac{\gamma}{2\eta} \mathbb{E}\left[\|w^k - \hat{u}^k\|^2\right]. \tag{9}$$

### 5.1 The Convex-Concave Case

For the convex-concave case, we use Jensen's inequality and the convexity (concavity) to bound the duality gap at $u_{\mathrm{avg}}^K = \frac{1}{K} \sum_{k=0}^{K-1} u^k$ as follows

$$\mathrm{Gap}(u_{\mathrm{avg}}^K) \le \max_{(x', y') \in \mathcal{Z}} \frac{1}{K} \sum_{k=0}^{K-1} \big( f(u_x^k, y') - f(x', u_y^k) \big) \le \max_{z \in \mathcal{Z}} \frac{1}{K} \sum_{k=0}^{K-1} \langle F(u^k), u^k - z \rangle. \tag{10}$$

Applying Lemma 1 by summing over inequality (9), we can bound the right-hand side of (10) via the terms of $\sum_{k=0}^{K-1} \mathbb{E}\left[\|z^k - \hat{u}^k\|^2\right]$ and $\sum_{k=0}^{K-1} \mathbb{E}\left[\|w^k - \hat{u}^k\|^2\right]$, and achieve the following theorem.

**Theorem 1.** *Suppose Assumptions 1, 2, 3, 4 and 6 hold with $0 < \delta \leq L$ and $D > 0$, we run Algorithm 1 with $b = \lceil\sqrt{n}\rceil$, $\gamma = p = 1/(\sqrt{n}+8)$, $\eta = \min\left\{\sqrt{\gamma b}/(4\delta), 1/(32\delta)\right\}$, $\alpha = 1$, and $\varepsilon_k = \min\left\{\zeta, \hat{c}^{-1}\min\left\{\|\hat{u}^k - z^k\|, \|\hat{u}^k - z^k\|^2\right\}\right\}$ for some $\zeta = \mathrm{poly}(L, \delta, n, D, \varepsilon)$ and $\hat{c} = \mathrm{poly}(\delta)$. Then we have*

$$\mathbb{E}\left[\max_{z \in \mathcal{Z}} \frac{1}{K}\sum_{k=0}^{K-1}\langle F(u^k), u^k - z\rangle\right] \leq \frac{10D^2}{\eta K} + \frac{\varepsilon}{2}, \quad \text{where } u_{\mathrm{avg}}^K = \frac{1}{K}\sum_{k=0}^{K-1} u^k.$$

Theorem 1 shows we can run SVOGS with step size $\eta = \Theta(1/\delta)$ and communication rounds of $K = \mathcal{O}(\delta D/\varepsilon)$ to achieve the $\varepsilon$-sub-optimality in expectation. Additionally, each communication round contains the expected communication complexity of $b(1 - p) + np = \mathcal{O}(\sqrt{n})$, leading to the overall communication complexity of $\mathcal{O}(n + \sqrt{n}\delta D^2/\varepsilon)$.

The sub-problem (2) in SVOGS (line 8 of Algorithm 1) is a minimax problem with $(L + 1/\eta)$-smooth and $(1/\eta)$-strongly-convex-$(1/\eta)$-strongly-concave objective. Therefore, the setting of $\varepsilon_k$ and $\eta$ in the theorem indicates the condition $\|u^k - \hat{u}^k\|^2 \leq \varepsilon_k$ can be achieved by the local iterations number of $\mathcal{O}((L + \delta)/\delta \log(\varepsilon_k)) = \tilde{\mathcal{O}}(L/\delta \log(1/\varepsilon))$ on the server (e.g., use EG [24]). Additionally, each round of SVOGS contains the expected local gradient complexity of $b(1 - p) + np = \mathcal{O}(\sqrt{n})$ to achieve the (mini-batch) optimistic gradient $\delta^k$. Hence, the overall local gradient complexity of SVOGS is $\tilde{\mathcal{O}}(K(\sqrt{n} + L/\delta \log(1/\varepsilon))) = \tilde{\mathcal{O}}(n + (\sqrt{n}\delta + L)D^2/\varepsilon \log(1/\varepsilon))$. We formally present the upper complexity bounds of SVOGS for the convex-concave case as follows.

**Corollary 1.** *Following the setting of Theorem 1, we can achieve $\mathbb{E}[\mathrm{Gap}(u_{\mathrm{avg}}^K)] \leq \varepsilon$ within communication rounds of $\mathcal{O}(\delta D^2/\varepsilon)$, communication complexity of $\mathcal{O}(n + \sqrt{n}\delta D^2/\varepsilon)$, and local gradient complexity of $\tilde{\mathcal{O}}(n + (\sqrt{n}\delta + L)D^2/\varepsilon \log(1/\varepsilon))$, where $u_{\mathrm{avg}}^K = \frac{1}{K}\sum_{k=0}^{K-1} u^k$.*

## 5.2 The Strongly-Convex-Strongly-Concave Case

By appropriate settings of SVOGS, Lemma 1 leads to the following linear convergence of our Lyapunov function in the strongly-convex-strongly-concave case.

**Theorem 2.** *Suppose Assumptions 1, 3, 4, 5 and 6 hold with $0 < \mu \leq \delta \leq L$, we run Algorithm 1 with $b = \lceil\min\{\sqrt{n}, \delta/\mu\}\rceil$, $\gamma = p = 1/(\min\{\sqrt{n}, \delta/\mu\} + 8)$, $\eta = \min\left\{\sqrt{\alpha\gamma b}/(4\delta), 1/(32\delta)\right\}$, $\alpha = \max\left\{1 - \eta\mu/(6(1 - \gamma)), 1 - p\eta\mu/(2\gamma + \eta\mu)\right\}$, and $\varepsilon_k = c^{-1}\min\left\{\|\hat{u}^k - z^k\|, \|\hat{u}^k - z^k\|^2\right\}$ for some $c = \mathrm{poly}(\mu, \delta)$. Then we have*

$$\mathbb{E}[\Phi^K] \leq \max\left\{1 - \frac{\eta\mu}{6(1 - \gamma)}, 1 - \frac{p\eta\mu}{2\gamma + \eta\mu}\right\}^K \Phi^0.$$

We then apply Theorem 2 with $K = \mathcal{O}(\delta/\mu \log(1/\varepsilon))$ and analyze the complexity like the discussion after Theorem 1, which results in the upper complexity bounds as follows.

**Corollary 2.** *Following the setting of Theorem 2, we can achieve $\mathbb{E}\left[\|z^K - z^*\|^2\right] \leq \varepsilon$ within communication rounds of $\mathcal{O}(\delta/\mu \log(1/\varepsilon))$, communication complexity of $\mathcal{O}((n + \sqrt{n}\delta/\mu)\log(1/\varepsilon))$, and local gradient complexity of $\tilde{\mathcal{O}}((n + (\sqrt{n}\delta + L)/\mu)\log(1/\varepsilon))$.*

## 5.3 Making the Gradient Mapping Small

For the convex-concave case (under the assumptions of Theorem 1), we can achieve the points with small gradient mapping by solving the regularized problem

$$\min_{x \in \mathcal{X}} \max_{y \in \mathcal{Y}} \hat{f}(x, y) := f(x, y) + \frac{\lambda}{2}\left\|x - x^0\right\|^2 - \frac{\lambda}{2}\left\|y - y^0\right\|^2 \tag{11}$$

for some $\lambda > 0$. Noticing that the function $\hat{f}(x, y)$ is $(L + \lambda)$-smooth, $\lambda$-strongly-convex-$\lambda$-strongly-concave and $\delta$-similarity. Then Corollary 2 implies running SVOGS (Algorithm 1) by iterations number $K = \mathcal{O}(\delta D/\sqrt{\varepsilon} \log(L/\varepsilon))$ to solve problem (11) with $\lambda = \mathcal{O}(\sqrt{\varepsilon}/D)$ can

achieve $\mathbb{E}[\|\mathscr{F}_\tau(z^K)\|^2] \leq \varepsilon$, which results in the complexity shown in Table 3. For the strongly-convex-strongly-concave case, the complexity of achieving $\mathbb{E}[\|\mathscr{F}_\tau(z^K)\|^2] \leq \varepsilon$ nearly matches the complexity of achieving $\mathbb{E}[\|z^K - z^*\|^2] \leq \varepsilon$. We defer the detailed derivation for these results of making the gradient mapping small to Appendix G.

## 6 The Optimality of SVOGS

In this section, we provide the lower complexity bounds for solving our minimax problems by using distributed first-order oracle (DFO) methods. The class of algorithms considered in our analysis follows the definition of Beznosikov et al. [8], which is formally described in Appendix D. Compared with existing lower bound analysis for second-order similarity only focusing on communication [5, 8], we additionally study the computation complexity by considering the local gradient calls. The results in this section imply the complexity of proposed SVOGS (nearly) matches the lower bounds on the communication rounds, the communication complexity and the local gradient calls simultaneously.

### 6.1 The Lower Bounds for Convex-Concave Case

We first provide the following lower bounds for convex-concave case.

**Theorem 3.** *For any $0 < \delta \leq L$, $n \geq 3$, $D > 0$ and $\varepsilon \leq \delta D^2/(12\sqrt{2})$, there exist $L$-smooth and convex-concave functions $f_1, \ldots, f_n : \mathbb{R}^{d_x} \times \mathbb{R}^{d_y}$ with $\delta$-second-order similarity, and closed convex set $\mathcal{Z} = \mathcal{X} \times \mathcal{Y}$ with diameter $D$. In order to find an approximate solution $z = (x, y)$ of problem (1) such that $\mathbb{E}[\mathrm{Gap}(z)] \leq \varepsilon$, any DFO algorithm needs at least $\Omega(\delta D^2/\varepsilon)$ communication rounds.*

**Theorem 4.** *For any $0 < \delta \leq L$, $n \geq 2$, $D > 0$ and $\varepsilon \leq \delta D^2/(16\sqrt{2n})$, there exist $L$-smooth and convex-concave functions $f_1, \ldots, f_n : \mathbb{R}^{d_x} \times \mathbb{R}^{d_y}$ with $\delta$-second-order similarity, and closed convex set $\mathcal{Z} = \mathcal{X} \times \mathcal{Y}$ with diameter $D$. In order to find an approximate solution $z = (x, y)$ of problem (1) such that $\mathbb{E}[\mathrm{Gap}(z)] \leq \varepsilon$, any DFO algorithm needs at least $\Omega(n + \sqrt{n}\delta D^2/\varepsilon)$ communication complexity and $\Omega(n + \sqrt{n}\delta D^2/\varepsilon)$ local gradient calls.*

The lower bounds on communication round and communication complexity shown in Theorem 3 and 4 match the corresponding upper bounds of SVOGS shown in Corollary 1. However, the lower bound on local gradient complexity shown in Theorem 4 only nearly matches the result of Corollary 1 in the case of $\sqrt{n}\delta \geq \Omega(L)$. Therefore, we also provide the following lower bound on local gradient complexity to show the tightness of dependence on the smoothness parameter $L$.

**Lemma 2.** *For any $L > 0$, $n \in \mathbb{N}$, $D > 0$ and $\varepsilon \leq \delta D^2/(4\sqrt{2})$, there exist $L$-smooth and convex-concave functions $f_1, \ldots, f_n : \mathbb{R}^{d_x} \times \mathbb{R}^{d_y}$ with $\delta$-second-order similarity, and closed convex set $\mathcal{Z} = \mathcal{X} \times \mathcal{Y}$ with diameter $D$. In order to find an approximate solution $z = (x, y)$ of problem (1) such that $\mathbb{E}[\mathrm{Gap}(z)] \leq \varepsilon$, any DFO algorithm needs at least $\Omega(n + LD^2/\varepsilon)$ local gradient calls.*

Combining the results of Theorem 4 and Lemma 2, we achieve the following lower bound on local gradient complexity, which nearly matches the corresponding upper bound shown in Corollary 1.

**Theorem 5.** *For any $0 < \delta \leq L$, $n \geq 2$, $D > 0$ and $\varepsilon \leq \delta D^2/(16\sqrt{2n})$, there exist $L$-smooth and convex-concave functions $f_1, \ldots, f_n : \mathbb{R}^{d_x} \times \mathbb{R}^{d_y}$ with $\delta$-second-order similarity, and closed convex set $\mathcal{Z} = \mathcal{X} \times \mathcal{Y}$ with diameter $D$. In order to find an approximate solution $z = (x, y)$ of problem (1) such that $\mathbb{E}[\mathrm{Gap}(z)] \leq \varepsilon$, any DFO algorithm needs at least $\Omega(n + (\sqrt{n}\delta + L)D^2/\varepsilon)$ local gradient calls.*

The constructions in our lower bound analysis is based on the modifications on the blinear functions provided by Han et al. [14], which are originally used to analyze the minimax optimization in non-distributed setting. We provide detailed proofs in Appendix E. In related work, Beznosikov et al. [5] also provide the lower bound of $\Omega(\delta D^2/\varepsilon)$ (matching the result of Theorem 3) for communication rounds by using the regularized function, which is different from our construction in the proof of Theorem 3. In addition, our lower bounds on the communication complexity and the local gradient complexity shown in Theorem 4 and 5 are new.

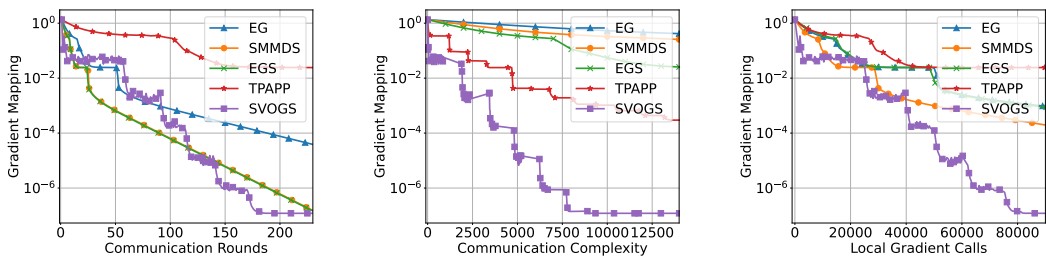

Figure 1: Results for convex-concave minimax problem (12) on a9a.

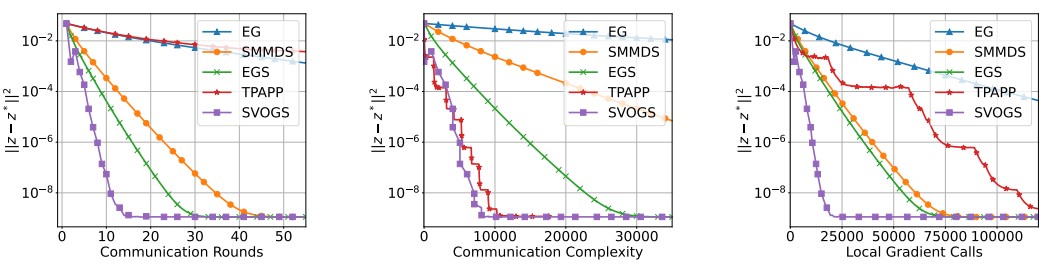

Figure 2: Results for strongly-convex-strongly-concave minimax problem (13) on a9a.

## 6.2 The Lower Bounds for Strongly-Convex-Strongly-Concave Case

The tight lower bound on communication rounds in strongly-convex-strongly-concave case has been provided by Beznosikov et al. [5, Theorem 1]. We present the result as follows.

**Theorem 6** ([5]). *For any $\mu, \delta, L > 0$ with $L \geq \max\{\mu, \delta\}$ and $n \geq 3$, there exist $L$-smooth and convex-concave functions $f_1, \ldots, f_n : \mathbb{R}^{d_x} \times \mathbb{R}^{d_y}$ with $\delta$-second-order similarity such that the function $f(x, y) = \frac{1}{n} \sum_{i=1}^{n} f_i(x, y)$ is $\mu$-strongly-convex-$\mu$-strongly-concave. In order to find a solution of problem (1) such that $\mathbb{E}[\|z - z^*\|^2] \leq \varepsilon$, any DFO algorithm needs at least $\Omega(\delta/\mu \log(1/\varepsilon))$ communication rounds.*

The tight lower bound on communication complexity has been provided by Beznosikov et al. [8]. We follow their construction to establish the lower bound on local gradient complexity, nearly matching the corresponding upper bound of our SVOGS. We formally present these lower bounds as follows.

**Theorem 7.** *For any $\mu, \delta, L > 0$ with $L \geq \max\{\mu, \delta\}$ and $n \geq 2$, there exist $L$-smooth and convex-concave functions $f_1, \ldots, f_n : \mathbb{R}^{d_x} \times \mathbb{R}^{d_y}$ with $\delta$-second-order similarity such that the function $f(x, y) = \frac{1}{n} \sum_{i=1}^{n} f_i(x, y)$ is $\mu$-strongly-convex-$\mu$-strongly-concave. In order to find a solution of problem (1) such that $\mathbb{E}[\|z - z^*\|^2] \leq \varepsilon$, any DFO algorithm needs at least $\Omega((n + \sqrt{n}\delta/\mu) \log(1/\varepsilon))$ communication complexity and $\Omega((n + (\sqrt{n}\delta + L)/\mu) \log(1/\varepsilon))$ local gradient calls.*

## 7 Experiments

We conduct the experiment on robust linear regression [5, 15, 35]. Concretely, we consider the constrained convex-concave minimax problem

$$\min_{\|x\|_1 \leq R_x} \max_{\|y\| \leq R_y} \frac{1}{2N} \sum_{i=1}^{N} \left( x^\top (a_i + y) - b_i \right)^2, \tag{12}$$

and the unconstrained strongly-convex-strongly-concave minimax problem

$$\min_{x \in \mathbb{R}^{d'}} \max_{y \in \mathbb{R}^{d'}} \frac{1}{2N} \sum_{i=1}^{N} \left( x^\top (a_i + y) - b_i \right)^2 + \frac{\lambda}{2} \|x\|^2 - \frac{\beta}{2} \|y\|^2, \tag{13}$$

where $x$ contains the weights of the model, $y$ describes the noise, and $\{(a_i, b_i)\}_{i=1}^{N}$ is the training set.

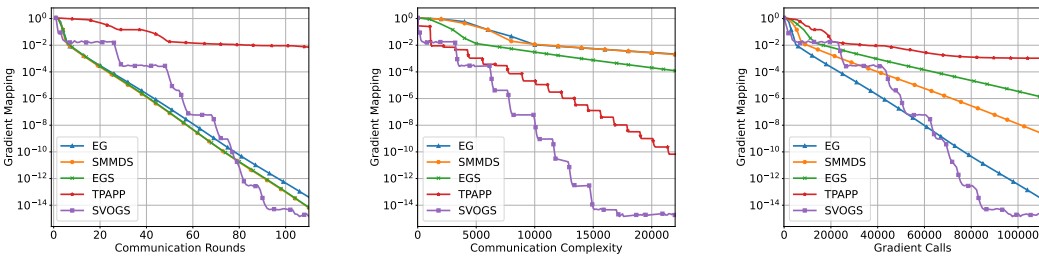

Figure 3: Results for convex-concave minimax problem (12) on w8a.

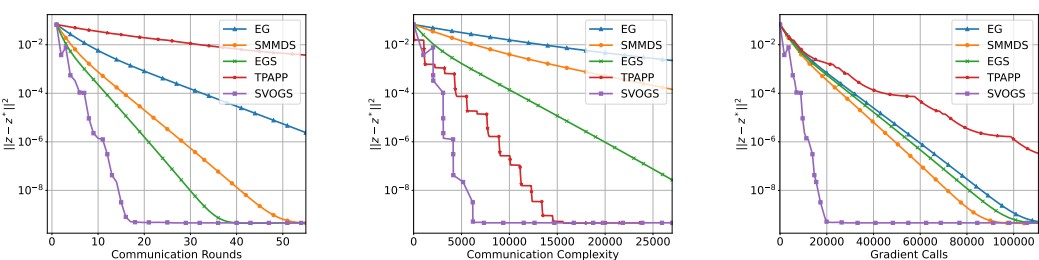

Figure 4: Results for strongly-convex-strongly-concave minimax problem (13) on w8a.

We compare the proposed SVOGS (Algorithm 1) with baselines Extra-Gradient method (EG) [24], Star Min-Max Data Similarity algorithm (SMMDS) [5], Extra-Gradient Sliding (EGS) [25]), and Three Pillars Algorithm with Partial Participation (TPAPP) [8]. We test the algorithms on real-world datasets "a9a" ($N = 32,561$, $d' = 123$), "w8a" ($N = 49,749$, $d' = 300$) and "covtype" ($N = 581,012$, $d' = 54$) from LIBSVM repository [9] and set the nodes number be $n = 500$. For problem (12), we set $R_x = 2$ and $R_y = 0.05$, respectively.

We implement all of the methods by Python 3.9 with NumPy and run on a machine with AMD Ryzen(TM) 7 4800H 8 core with Radeon Graphics 2.90 GHz CPU with 16GB RAM. We solve the sub-problem in SVOGS (Algorithm 1), SMMDS [5], EGS [25], and TPAPP [8] by Extra-Gradient method of Korpelevich [24]. We tune the step-size $\eta$ of SVOGS from $\{0.01, 0.1, 1\}$. The probability $p$ is tuned from $\{p_0, 5p_0, 10p_0\}$, where $p_0 = 1/\min\{\sqrt{n}+\delta/\mu\}$. The batch size $b$ is determined from $\{\lfloor b_0/10\rfloor, \lfloor b_0/5\rfloor, \lfloor b_0\rfloor\}$, with $b_0 = 1/p_0$. We set the other parameters by following our theoretical analysis. We set the average weight as $\gamma = 1 - p$. For the momentum parameter, we set $\alpha = 1$ for convex-concave case and $\alpha = \max\{1 - \eta\mu/(6(1 - \gamma)), 1 - p\eta\mu/(2\gamma + \eta\mu)\}$ for strongly-convex-strongly-concave case, where we estimate $\mu$ by $\max\{\lambda, \beta\}$ for problem (13). For the sub-problem solver, we set its step-size according to the smoothness parameter of sub-problem, i.e., $1/(L + 1/\eta)$. In addition, we estimate the smooth parameter $L$ and the similarity parameter $\delta$ by following the strategy in Appendix C of Beznosikov et al. [5].

We present the experimental results in Figure 1 to 4 for datasets "a9a" and "w8a". The results for dataset "covtype" is displayed in Appendix H due to the space limitation. We can observe that our SVOGS outperforms all baselines in terms of the local gradient complexity. Additionally, the SVOGS requires less communication rounds than classical EG and existing partial participation method TPAPP, and it requires significantly less communication complexity than full participation methods EG, SMMDS and EGS. All of these empirical results support our theoretical analysis.

## 8 Conclusion

This paper presents a novel distributed optimization method named SVOGS, which use the second-order similarity in local functions and the finite-sum structure in objective to solve the convex-concave minimax problem within the near-optimal complexity. Our theoretical results are also validated by the numerical experiments. In future work, it is interesting to use our ideas to improve the efficiency of distributed nonconvex minimax optimization under the second-order similarity.

## Acknowledgments and Disclosure of Funding

Luo and Zhou is supported by the Major Key Project of Pengcheng Laboratory (No. PCL2024A06), National Natural Science Foundation of China (No. 62206058), Shanghai Sailing Program (22YF1402900), and Shanghai Basic Research Program (23JC1401000). Ye is supported in part by the National Natural Science Foundation of China under Grant 12101491 and in part by the National Key Research and Development Project of China under Grant 2022YFA1004002.

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

# Appendix

The appendix contains additional details supporting the main text. Section A starts with some basic results. Section B shows the non-negativity of our Lyapunov function. Section C provides the proof of upper bounds for the proposed method. Section D formally defines the algorithm class in our lower bound analysis. Section E and F provide the lower complexity bound for both convex-concave and strongly-convex-strongly-concave cases. Section G demonstrates the complexity of making the gradient mapping small. Section H presents more experimental results.

## A Some Basic Results

We introduce the following lemmas for our later analysis.

**Lemma 3** (Lin et al. [29, Proposition B.1]). *If the local functions $f_1, \ldots, f_n : \mathbb{R}^{d_x} \times \mathbb{R}^{d_y} \to \mathbb{R}$ hold the $\delta$-second-order similarity, then each $(F_i - F)(\cdot)$ is $\delta$-Lipschitz continuous, i.e., we have*

$$\|(F_i - F)(z_1) - (F_i - F)(z_2)\| \le \delta \|z_1 - z_2\|$$

*for all $z_1, z_2 \in \mathbb{R}^d$ and $i \in [n]$.*

**Lemma 4** (Alacaoglu and Malitsky [1, Section 8]). *Let $\mathcal{F} = \{\mathcal{F}_k\}_{k \ge 0}$ be a filtration and $\{r^k\}$ be a stochastic process adapted to $\mathcal{F}$ with $\mathbb{E}[r^{k+1} | \mathcal{F}_k] = 0$. Then for any $K \in \mathbb{N}$, $x^0 \in \mathcal{Z}$, and any compact set $\mathcal{C} \subset \mathcal{Z}$, we have*

$$\mathbb{E}\left[\max_{x \in \mathcal{C}} \sum_{k=0}^{K-1} \langle r^{k+1}, x \rangle\right] \le \max_{x \in \mathcal{C}} \frac{1}{2}\|x^0 - x\|^2 + \frac{1}{2}\sum_{k=0}^{K-1} \mathbb{E}\|r^{k+1}\|^2.$$

In related work, Beznosikov et al. [8, Assumption 4.3] considers the following second-order similarity assumption that is slightly different from our Assumption 6.

**Assumption 7.** *The local functions $f_1, \ldots, f_n : \mathbb{R}^{d_x} \times \mathbb{R}^{d_y} \to \mathbb{R}$ are twice differentiable and hold the $\delta$-average-second-order similarity, i.e., there exists $\delta > 0$ such that*

$$\frac{1}{n}\sum_{i=1}^{n} \left\|\nabla^2 f_i(x,y) - \nabla^2 f_j(x,y)\right\|^2 \le \delta^2$$

*for all $x \in \mathbb{R}^{d_x}$, $y \in \mathbb{R}^{d_y}$, and $j \in [n]$, .*

We present the relationship between $\delta$-average-second-order similarity (Assumption 7) and $\delta$-second-order-similarity (Assumption 6).

**Proposition 1.** *For twice differentiable local functions $f_1, \ldots, f_n : \mathbb{R}^{d_x} \times \mathbb{R}^{d_y} \to \mathbb{R}$, we have*

- *If functions $\{f_i\}_{i=1}^{n}$ hold the $\delta$-average-second-order similarity, then they also hold the $\delta$-second-order similarity.*
- *If functions $\{f_i\}_{i=1}^{n}$ hold the $\delta$-second-order similarity, then they hold the $2\delta$-average-second-order similarity.*

*Proof.* If functions $\{f_i\}_{i=1}^{n}$ hold the $\delta$-average-second-order similarity, then for all $j \in [n]$, we have

$$\|\nabla^2 f_j(x,y) - \nabla^2 f(x,y)\|_2^2 = \left\|\frac{1}{n}\sum_{i=1}^{n}\left[\nabla^2 f_j(x,y) - \nabla^2 f_i(x,y)\right]\right\|_2^2$$

$$\le \frac{1}{n}\sum_{i=1}^{n}\|\nabla^2 f_j(x,y) - \nabla^2 f_i(x,y)\|_2^2 \le \delta^2,$$

where we use the convexity of $\|\cdot\|_2^2$. This implies functions $\{f_i\}_{i=1}^{n}$ also hold the $\delta$-second-order similarity.

If functions $\{f_i\}_{i=1}^{n}$ hold the $\delta$ second-order similarity, then for all $j \in [n]$, we have

$$\frac{1}{n}\sum_{i=1}^{n}\|\nabla^2 f_j(x,y) - \nabla^2 f_i(x,y)\|_2^2$$

$$\le \frac{1}{n}\sum_{i=1}^{n}\left(2\|\nabla^2 f_j(x,y) - \nabla^2 f(x,y)\|_2^2 + 2\|\nabla^2 f(x,y) - \nabla^2 f_i(x,y)\|_2^2\right) \le (2\delta)^2,$$

where we use the Young's inequality for the matrix 2-norm. This implies functions $\{f_i\}_{i=1}^n$ hold the $2\delta$-average-second-order similarity. $\qquad\square$

## B  The Non-Negativity of Lyapunov Function

Our convergence analysis is based on the following Lyapunov function

$$\Phi^k = \left(\frac{1-\gamma}{\eta} + \frac{\mu}{2}\right)\|z^k - z^*\|^2 + 2\langle F(z^{k-1}) - F_1(z^{k-1}) - F(z^k) + F_1(z^k), z^k - z^*\rangle$$

$$+ \frac{1}{64\eta}\|z^k - z^{k-1}\|^2 + \frac{\gamma}{2\eta}\|w^{k-1} - z^k\|^2 + \frac{(2\gamma + \eta\mu)}{2p\eta}\|w^k - z^*\|^2.$$

Noticing that we can always guarantees $\Phi^k \geq 0$ by taking $\eta \leq 1/(32\delta)$, because the Young's inequality and Lemma 3 indicates

$$\Phi^k \geq \frac{1-\gamma}{\eta}\|z^k - z^*\|^2 - \frac{1}{64\eta\delta^2}\|F(z^{k-1}) - F_1(z^{k-1}) - F(z^k) + F_1(z^k)\|^2$$

$$- 64\eta\delta^2\|z^k - z^*\|^2 + \frac{1}{64\eta}\|z^k - z^{k-1}\|^2$$

$$\geq \frac{1-\gamma}{\eta}\|z^k - z^*\|^2 - \frac{1}{64\eta\delta^2}\delta^2\|z^k - z^{k-1}\|^2 - 64\eta\delta^2\|z^k - z^*\|^2 + \frac{1}{64\eta}\|z^k - z^{k-1}\|^2$$

$$\geq \frac{1}{2\eta}(1 - 128\eta^2\delta^2)\|z^k - z^*\|^2 \geq 0,$$

where we also use $\gamma \leq 1/8$.

## C  The Proofs for Upper Complexity Bounds

We provide the proofs for results in Section 5.

### C.1  Proof of Lemma 1

In later analysis, we denote $\mathbb{E}_k[\cdot]$ as the expectation with respect to the random sampled set $\mathcal{S}^k$ in round $k$ and denote $\mathbb{E}_{k+1/2}[\cdot]$ as the expectation with respect to the random update of the snapshot point $w^k$ in round $k$. Specifically, we take the constant

$$c := 100 + \frac{64\eta\delta^2}{\mu} + 2048\eta^2\delta^2 + 96\eta\mu + 64\sqrt{2\eta\Phi^0} \tag{14}$$

for the statement of Lemma 1.

We first provide several lemmas that will be used in the proof of Lemma 1.

**Lemma 5.** *Under the setting of Lemma 1, we have*

$$- 2\mathbb{E}\left[\langle \delta^k - \mathbb{E}_k[\delta^k], \hat{u}^k - z^k\rangle\right]$$

$$\leq \frac{1}{2\eta}\mathbb{E}\left[\|\hat{u}^k - z^k\|^2\right] + \frac{4\eta\delta^2}{b}\mathbb{E}\left[\|z^k - w^{k-1}\|^2\right] + \frac{4\eta\delta^2\alpha^2}{b}\mathbb{E}\left[\|z^k - z^{k-1}\|^2\right],$$

*and*

$$- 2\mathbb{E}\left[\langle \mathbb{E}_k[\delta^k] + F_1(\hat{u}^k) - F(z^*), \hat{u}^k - z^*\rangle\right]$$

$$\leq \frac{2\delta^2}{\mu}\mathbb{E}\left[\|\hat{u}^k - u^k\|^2\right] + \frac{\mu}{2}\mathbb{E}\left[\|\hat{u}^k - z^*\|^2\right] - 2\mu\mathbb{E}\left[\|\hat{u}^k - z^*\|^2\right] + \frac{1}{64\eta}\mathbb{E}\left[\|z^k - z^{k+1}\|^2\right]$$

$$+ 64\eta\delta^2\mathbb{E}\left[\|\hat{u}^k - u^k\|^2\right] - 2\mathbb{E}\left[\langle F(z^k) - F_1(z^k) - F(z^{k+1}) + F_1(z^{k+1}), z^{k+1} - z^*\rangle\right]$$

$$+ \frac{1}{4\eta}\mathbb{E}\left[\|z^k - \hat{u}^k\|^2\right] + 4\eta\delta^2\alpha^2\mathbb{E}\left[\|z^k - z^{k-1}\|^2\right]$$

$$- 2\alpha\mathbb{E}\left[\langle F(z^k) - F_1(z^k) - F(z^{k-1}) + F_1(z^{k-1}), z^k - z^*\rangle\right].$$

*Proof.* Firstly note that
$$\mathbb{E}_k[\delta^k] = F(z^k) - F_1(z^k) + \alpha\left(F(z^k) - F_1(z^k) - F(z^{k-1}) + F_1(z^{k-1})\right).$$
According to the uniform and independent sampling and Lemma 3 we have

$$\mathbb{E}\left[\|\delta^k - \mathbb{E}_k[\delta^k]\|^2\right] \leq 2\mathbb{E}\left\|\frac{1}{b}\sum_{j\in\mathcal{S}^k}\left(F_j(z^k) - F_j(w^{k-1})\right) - \left(F(z^k) - F(w^{k-1})\right)\right\|^2$$

$$+ 2\mathbb{E}\left\|\frac{\alpha}{b}\sum_{j\in\mathcal{S}^k}\left(F_j(z^k) - F_j(z^{k-1})\right) - \left(F(z^k) - F(z^{k-1})\right)\right\|^2$$

$$= \frac{2}{b^2}\mathbb{E}\left[\sum_{j\in\mathcal{S}^k}\left\|\left(F_j(z^k) - F_j(w^{k-1})\right) - \left(F(z^k) - F(w^{k-1})\right)\right\|^2\right]$$

$$+ \frac{2\alpha^2}{b^2}\mathbb{E}\left[\sum_{j\in\mathcal{S}^k}\left\|\left(F_j(z^k) - F_j(z^{k-1})\right) - \left(F(z^k) - F(z^{k-1})\right)\right\|^2\right]$$

$$\leq \frac{2}{nb}\mathbb{E}\left[\sum_{j=1}^n\left\|\left(F_j(z^k) - F_j(w^{k-1})\right) - \left(F(z^k) - F(w^{k-1})\right)\right\|^2\right]$$

$$+ \frac{2\alpha^2}{nb}\mathbb{E}\left[\sum_{j=1}^n\left\|\left(F_j(z^k) - F_j(z^{k-1})\right) - \left(F(z^k) - F(z^{k-1})\right)\right\|^2\right]$$

$$\leq \frac{2\delta^2}{b}\mathbb{E}\left[\|z^k - w^{k-1}\|^2\right] + \frac{2\delta^2\alpha^2}{b}\mathbb{E}\left[\|z^k - z^{k-1}\|^2\right].$$

According to the above bound on $\mathbb{E}\left[\|\delta^k - \mathbb{E}_k[\delta^k]\|^2\right]$, we achieve the first result as follows

$$-2\mathbb{E}\left[\langle\delta^k - \mathbb{E}_k[\delta^k], \hat{u}^k - z^k\rangle\right]$$

$$\leq \frac{1}{2\eta}\mathbb{E}\left[\|\hat{u}^k - z^k\|^2\right] + 2\eta\mathbb{E}\left[\|\delta^k - \mathbb{E}_k[\delta^k]\|^2\right]$$

$$\leq \frac{1}{2\eta}\mathbb{E}\left[\|\hat{u}^k - z^k\|^2\right] + \frac{4\eta\delta^2}{b}\mathbb{E}\left[\|z^k - w^{k-1}\|^2\right] + \frac{4\eta\delta^2\alpha^2}{b}\mathbb{E}\left[\|z^k - z^{k-1}\|^2\right].$$

Again using Lemma 3, we achieve the second result as follows

$$-2\mathbb{E}\left[\langle\mathbb{E}_k[\delta^k] + F_1(\hat{u}^k) - F(z^*), \hat{u}^k - z^*\rangle\right]$$

$$= -2\mathbb{E}\left[\langle F(z^k) - F_1(z^k) + \alpha\left(F(z^k) - F_1(z^k) - F(z^{k-1}) + F_1(z^{k-1})\right), \hat{u}^k - z^*\rangle\right]$$

$$-2\mathbb{E}\left[\langle F_1(\hat{u}^k) - F(z^*), \hat{u}^k - z^*\rangle\right]$$

$$= -2\mathbb{E}\left[\langle F(u^k) - F_1(u^k) - F(\hat{u}^k) + F_1(\hat{u}^k), \hat{u}^k - z^*\rangle\right] - 2\mathbb{E}\left[\langle F(\hat{u}^k) - F(z^*), \hat{u}^k - z^*\rangle\right]$$

$$-2\mathbb{E}\left[\langle F(z^k) - F_1(z^k) - F(z^{k+1}) + F_1(z^{k+1}), \hat{u}^k - z^{k+1}\rangle\right]$$

$$-2\mathbb{E}\left[\langle F(z^k) - F_1(z^k) - F(z^{k+1}) + F_1(z^{k+1}), z^{k+1} - z^*\rangle\right]$$

$$-2\alpha\mathbb{E}\left[\langle F(z^k) - F_1(z^k) - F(z^{k-1}) + F_1(z^{k-1}), \hat{u}^k - z^k\rangle\right]$$

$$-2\alpha\mathbb{E}\left[\langle F(z^k) - F_1(z^k) - F(z^{k-1}) + F_1(z^{k-1}), z^k - z^*\rangle\right]$$

$$\leq \frac{2\delta^2}{\mu}\mathbb{E}\left[\|\hat{u}^k - u^k\|^2\right] + \frac{\mu}{2}\mathbb{E}\left[\|\hat{u}^k - z^*\|^2\right] - 2\mu\mathbb{E}\left[\|\hat{u}^k - z^*\|^2\right] + \frac{1}{64\eta}\mathbb{E}\left[\|z^k - z^{k+1}\|^2\right]$$

$$+ 64\eta\delta^2\mathbb{E}\left[\|\hat{u}^k - u^k\|^2\right] - 2\mathbb{E}\left[\langle F(z^k) - F_1(z^k) - F(z^{k+1}) + F_1(z^{k+1}), z^{k+1} - z^*\rangle\right]$$

$$+ \frac{1}{4\eta}\mathbb{E}\left[\|z^k - \hat{u}^k\|^2\right] + 4\eta\delta^2\alpha^2\mathbb{E}\left[\|z^k - z^{k-1}\|^2\right]$$

$$-2\alpha\mathbb{E}\left[\langle F(z^k) - F_1(z^k) - F(z^{k-1}) + F_1(z^{k-1}), z^k - z^*\rangle\right].$$

$\square$

**Lemma 6.** *Under setting of Lemma 1, we have*

$$-\frac{1}{8\eta}\mathbb{E}\left[\|\hat{u}^k - z^k\|^2\right] - \frac{\gamma}{\eta}\mathbb{E}\left[\|w^k - \hat{u}^k\|^2\right] - \frac{3\mu}{2}\mathbb{E}\left[\|\hat{u}^k - z^*\|^2\right]$$

$$\leq -\frac{1}{16\eta}\mathbb{E}\left[\|\hat{u}^k - z^k\|^2\right] - \frac{1}{32\eta}\mathbb{E}\left[\|z^{k+1} - z^k\|^2\right] + \left(\frac{1}{8\eta} + 3\mu\right)\mathbb{E}\left[\|\hat{u}^k - u^k\|^2\right]$$

$$- \frac{\gamma}{2\eta}\mathbb{E}\left[\|w^k - \hat{u}^k\|^2\right] - \frac{\gamma}{4\eta}\mathbb{E}\left[\|w^k - z^{k+1}\|^2\right] - \mu\mathbb{E}\left[\|z^{k+1} - z^*\|^2\right].$$

*Proof.* From the facts $\|a + b\|^2 \geq \frac{1}{2}\|a\|^2 - \|b\|^2$ and $\frac{3}{2}\|a + b\|^2 \geq \|a\|^2 - 3\|b\|^2$, we have

$$-\frac{1}{8\eta}\mathbb{E}\left[\|\hat{u}^k - z^k\|^2\right] \leq -\frac{1}{16\eta}\mathbb{E}\left[\|\hat{u}^k - z^k\|^2\right] - \frac{1}{32\eta}\mathbb{E}\left[\|z^{k+1} - z^k\|^2\right] + \frac{1}{16\eta}\mathbb{E}\left[\|\hat{u}^k - u^k\|^2\right],$$

$$-\frac{\gamma}{\eta}\mathbb{E}\left[\|w^k - \hat{u}^k\|^2\right] \leq -\frac{\gamma}{2\eta}\mathbb{E}\left[\|w^k - \hat{u}^k\|^2\right] - \frac{\gamma}{4\eta}\mathbb{E}\left[\|w^k - z^{k+1}\|^2\right] + \frac{\gamma}{2\eta}\mathbb{E}\left[\|\hat{u}^k - u^k\|^2\right]$$

$$\leq -\frac{\gamma}{2\eta}\mathbb{E}\left[\|w^k - \hat{u}^k\|^2\right] - \frac{\gamma}{4\eta}\mathbb{E}\left[\|w^k - z^{k+1}\|^2\right] + \frac{1}{16\eta}\mathbb{E}\left[\|\hat{u}^k - u^k\|^2\right],$$

and

$$-\frac{3\mu}{2}\mathbb{E}\left[\|\hat{u}^k - z^*\|^2\right] \leq -\mu\mathbb{E}\left[\|z^{k+1} - z^*\|^2\right] + 3\mu\mathbb{E}\left[\|\hat{u}^k - u^k\|^2\right],$$

where we use the setting $\gamma \leq 1/8$ in the second inequality. $\qquad\square$

**Lemma 7.** *Under setting of Lemma 1, we have*

$$\left(\frac{1-\gamma}{\eta} + \frac{\mu}{2}\right)\mathbb{E}\left[\|z^{k+1} - z^*\|^2\right] + 2\mathbb{E}\left[\langle F(z^k) - F_1(z^k) - F(z^{k+1}) + F_1(z^{k+1}), z^{k+1} - z^*\rangle\right]$$

$$+ \frac{1}{64\eta}\mathbb{E}\left[\|z^{k+1} - z^k\|^2\right] + \frac{\gamma}{4\eta}\mathbb{E}\left[\|w^k - z^{k+1}\|^2\right] + \frac{\gamma + \frac{1}{2}\eta\mu}{p\eta}\mathbb{E}\left[\|w^{k+1} - z^*\|^2\right]$$

$$\leq \frac{1-\gamma}{\eta}\mathbb{E}\left[\|z^k - z^*\|^2\right] + 2\alpha\mathbb{E}\left[\langle F(z^{k-1}) - F_1(z^{k-1}) - F(z^k) + F_1(z^k), z^k - z^*\rangle\right]$$

$$+ \alpha\frac{1}{64\eta}\mathbb{E}\left[\|z^k - z^{k-1}\|^2\right] + \frac{4\eta\delta^2}{b}\mathbb{E}\left[\|w^{k-1} - z^k\|^2\right] - \frac{1}{16\eta}\mathbb{E}\left[\|z^k - \hat{u}^k\|^2\right]$$

$$+ \left(-\frac{7}{8\eta} + \frac{2\delta^2}{\mu} + 64\eta\delta^2 + 3\mu\right)\mathbb{E}\left[\|\hat{u}^k - u^k\|^2\right] + \frac{2}{\eta}\mathbb{E}\left[\|\hat{u}^k - u^k\|\|z^{k+1} - z^*\|\right]$$

$$+ \left(1 - \frac{p\eta\mu}{2\gamma + \eta\mu}\right)\frac{(\gamma + \frac{1}{2}\eta\mu)}{p\eta}\mathbb{E}\left[\|w^k - z^*\|^2\right] - \frac{\gamma}{2\eta}\mathbb{E}\left[\|w^k - \hat{u}^k\|^2\right].$$

*Proof.* The optimality of $\hat{u}^k$ implies

$$\langle \eta F_1(\hat{u}^k) + \hat{u}^k - v^k, z^* - \hat{u}^k\rangle \geq 0. \tag{15}$$

Combine equation (15) with the update rule in Line 7 of Algorithm 1 and $\langle -\gamma F(z^*), z^* - \hat{u}^k\rangle \geq 0$, we achieve

$$-\frac{1}{\eta}\langle \bar{z}^k - \hat{u}^k - \eta\delta^k, \hat{u}^k - z^*\rangle \leq -\langle F_1(\hat{u}^k) - F(z^*), \hat{u}^k - z^*\rangle. \tag{16}$$

Using the result of equation (16), we have

$$
\begin{aligned}
\frac{1}{\eta}\|\hat{u}^k - z^*\|^2 &= \frac{1}{\eta}\|z^k - z^*\|^2 + \frac{2}{\eta}\langle \hat{u}^k - z^k, \hat{u}^k - z^*\rangle - \frac{1}{\eta}\|\hat{u}^k - z^k\|^2 \\
&= \frac{1}{\eta}\|z^k - z^*\|^2 + \frac{2\gamma}{\eta}\langle w^k - z^k, \hat{u}^k - z^*\rangle - 2\langle \delta^k, \hat{u}^k - z^*\rangle \\
&\quad - \frac{1}{\eta}\|\hat{u}^k - z^k\|^2 - \frac{2}{\eta}\langle \bar{z}^k - \hat{u}^k - \eta\delta^k, \hat{u}^k - z^*\rangle \\
&\leq \frac{1}{\eta}\|z^k - z^*\|^2 + \frac{2\gamma}{\eta}\langle w^k - z^k, \hat{u}^k - z^*\rangle - 2\langle \delta^k, \hat{u}^k - z^*\rangle \\
&\quad - \frac{1}{\eta}\|\hat{u}^k - z^k\|^2 - 2\langle F_1(\hat{u}^k) - F(z^*), \hat{u}^k - z^*\rangle \\
&= \frac{1}{\eta}\|z^k - z^*\|^2 + \frac{\gamma}{\eta}\|w^k - z^*\|^2 - \frac{\gamma}{\eta}\|w^k - \hat{u}^k\|^2 - \frac{\gamma}{\eta}\|z^k - z^*\|^2 \\
&\quad - \frac{1-\gamma}{\eta}\|\hat{u}^k - z^k\|^2 - 2\langle \delta^k - \mathbb{E}_k[\delta^k], \hat{u}^k - z^k + z^k - z^*\rangle \\
&\quad - 2\langle \mathbb{E}_k[\delta^k] + F_1(\hat{u}^k) - F(z^*), \hat{u}^k - z^*\rangle.
\end{aligned}
$$

Taking the expectation on above result and using the fact

$$
\mathbb{E}\left[\langle \delta^k - \mathbb{E}_k[\delta^k], z^k - z^*\rangle\right] = \mathbb{E}\left[\mathbb{E}_k\left[\langle \delta^k - \mathbb{E}_k[\delta^k], z^k - z^*\rangle\right]\right] = 0,
$$

we obtain

$$
\begin{aligned}
\frac{1}{\eta}\mathbb{E}\left[\|\hat{u}^k - z^*\|^2\right] &\leq \mathbb{E}\left[\frac{1}{\eta}\|z^k - z^*\|^2 + \frac{\gamma}{\eta}\|w^k - z^*\|^2 - \frac{\gamma}{\eta}\|w^k - \hat{u}^k\|^2 - \frac{\gamma}{\eta}\|z^k - z^*\|^2\right] \\
&\quad - \mathbb{E}\left[\frac{1-\gamma}{\eta}\|\hat{u}^k - z^k\|^2\right] - 2\mathbb{E}\left[\langle \delta^k - \mathbb{E}_k[\delta^k], \hat{u}^k - z^k\rangle\right] \\
&\quad - 2\mathbb{E}\left[\langle \mathbb{E}_k[\delta^k] + F_1(\hat{u}^k) - F(z^*), \hat{u}^k - z^*\rangle\right].
\end{aligned}
\tag{17}
$$

Applying Lemma 5 to bound the term $-2\mathbb{E}\left[\langle \delta^k - \mathbb{E}_k[\delta^k], \hat{u}^k - z^k\rangle\right]$ in equation (17), we obtain

$$
\begin{aligned}
&\frac{1}{\eta}\mathbb{E}\left[\|\hat{u}^k - z^*\|^2\right] \\
&\leq \mathbb{E}\left[\frac{1}{\eta}\|z^k - z^*\|^2 + \frac{\gamma}{\eta}\|w^k - z^*\|^2 - \frac{\gamma}{\eta}\|w^k - \hat{u}^k\|^2 - \frac{\gamma}{\eta}\|z^k - z^*\|^2 - \frac{1/4 - \gamma}{\eta}\|\hat{u}^k - z^k\|^2\right] \\
&\quad + \frac{4\eta\delta^2}{b}\mathbb{E}\left[\|z^k - w^{k-1}\|^2\right] + \frac{4\eta\delta^2\alpha^2}{b}\mathbb{E}\left[\|z^k - z^{k-1}\|^2\right] \\
&\quad + \frac{2\delta^2}{\mu}\mathbb{E}\left[\|\hat{u}^k - u^k\|^2\right] - \frac{3}{2}\mu\mathbb{E}\left[\|\hat{u}^k - z^*\|^2\right] + \frac{1}{64\eta}\mathbb{E}\left[\|z^k - z^{k+1}\|^2\right] + 64\eta\delta^2\mathbb{E}\left[\|\hat{u}^k - u^k\|^2\right] \\
&\quad - 2\mathbb{E}\left[\langle F(z^k) - F_1(z^k) - F(z^{k+1}) + F_1(z^{k+1}), z^{k+1} - z^*\rangle\right] + 4\eta\delta^2\alpha^2\mathbb{E}\left[\|z^k - z^{k-1}\|^2\right] \\
&\quad - 2\alpha\mathbb{E}\left[\langle F(z^k) - F_1(z^k) - F(z^{k-1}) + F_1(z^{k-1}), z^k - z^*\rangle\right] \\
&\leq \frac{1}{\eta}\mathbb{E}\left[\|z^k - z^*\|^2\right] + \frac{\gamma}{\eta}\mathbb{E}\left[\|w^k - z^*\|^2\right] - \frac{\gamma}{\eta}\mathbb{E}\left[\|w^k - \hat{u}^k\|^2\right] - \frac{\gamma}{\eta}\mathbb{E}\left[\|z^k - z^*\|^2\right] \\
&\quad - \frac{1}{8\eta}\mathbb{E}\left[\|\hat{u}^k - z^k\|^2\right] + \frac{1}{64\eta}\mathbb{E}\left[\|z^k - z^{k+1}\|^2\right] + \frac{4\eta\delta^2}{b}\mathbb{E}\left[\|z^k - w^{k-1}\|^2\right] \\
&\quad + \left(\frac{4\eta\delta^2\alpha^2}{b} + 4\eta\delta^2\alpha^2\right)\mathbb{E}\left[\|z^k - z^{k-1}\|^2\right] + \left(\frac{2\delta^2}{\mu} + 64\eta\delta^2\right)\mathbb{E}\left[\|\hat{u}^k - u^k\|^2\right] \\
&\quad - 2\mathbb{E}\left[\langle F(z^k) - F_1(z^k) - F(z^{k+1}) + F_1(z^{k+1}), z^{k+1} - z^*\rangle\right] - \frac{3\mu}{2}\mathbb{E}\left[\|\hat{u}^k - z^*\|^2\right] \\
&\quad - 2\alpha\mathbb{E}\left[\langle F(z^k) - F_1(z^k) - F(z^{k-1}) + F_1(z^{k-1}), z^k - z^*\rangle\right],
\end{aligned}
\tag{18}
$$

where we use the setting $\gamma \leq 1/8$ in the second inequality.

Then we consider the terms related to $\hat{u}$. Firstly, we have

$$\frac{1}{\eta}\mathbb{E}\left[\|\hat{u}^k - z^*\|^2\right] = \frac{1}{\eta}\mathbb{E}\left[\|z^{k+1} - z^*\|^2\right] + \frac{1}{\eta}\mathbb{E}\left[\|\hat{u}^k - u^k\|^2\right] - \frac{2}{\eta}\mathbb{E}\left[\|\hat{u}^k - u^k\|\|z^{k+1} - z^*\|\right].$$
(19)

Applying Lemma 6 and plugging equation (19) into equation (18), we have

$$\frac{1}{\eta}\mathbb{E}\left[\|z^{k+1} - z^*\|^2\right] + 2\mathbb{E}\left[\langle F(z^k) - F_1(z^k) - F(z^{k+1}) + F_1(z^{k+1}), z^{k+1} - z^*\rangle\right]$$

$$+ \frac{1}{64\eta}\mathbb{E}\left[\|z^{k+1} - z^k\|^2\right] + \frac{\gamma}{4\eta}\mathbb{E}\left[\|w^k - z^{k+1}\|^2\right]$$

$$\leq \frac{1}{\eta}\mathbb{E}\left[\|z^k - z^*\|^2\right] + 2\alpha\mathbb{E}\left[\langle F(z^{k-1}) - F_1(z^{k-1}) - F(z^k) + F_1(z^k), z^k - z^*\rangle\right]$$

$$+ \frac{\alpha}{64\eta}\mathbb{E}\left[\|z^k - z^{k-1}\|^2\right] + \frac{\gamma}{\eta}\mathbb{E}\left[\|w^k - z^*\|^2\right] - \frac{\gamma}{\eta}\mathbb{E}\left[\|z^k - z^*\|^2\right] - \mu\mathbb{E}\left[\|z^{k+1} - z^*\|^2\right]$$

$$+ \frac{4\eta\delta^2}{b}\mathbb{E}\left[\|z^k - w^{k-1}\|^2\right] + \frac{2}{\eta}\mathbb{E}\left[\|\hat{u}^k - u^k\|\|z^{k+1} - z^*\|\right] - \frac{1}{16\eta}\mathbb{E}\left[\|\hat{u}^k - z^k\|^2\right]$$

$$+ \left(-\frac{7}{8\eta} + \frac{2\delta^2}{\mu} + 64\eta\delta^2 + 3\mu\right)\mathbb{E}\left[\|\hat{u}^k - u^k\|^2\right] - \frac{\gamma}{2\eta}\mathbb{E}\left[\|w^k - \hat{u}^k\|^2\right],$$
(20)

where we use the fact that $256\eta^2\delta^2\alpha^2/b + 256\eta^2\delta^2\alpha^2 \leq \alpha$ to bound the coefficient before the term of $\mathbb{E}\left[\|z^k - z^{k-1}\|^2\right]$.

Then we add the term

$$\mu\mathbb{E}\left[\|z^{k+1} - z^*\|^2\right] + \frac{\gamma + \frac{1}{2}\eta\mu}{p\eta}\mathbb{E}\left[\|w^{k+1} - z^*\|^2\right]$$
(21)

to both sides of equation (20) and use the update rule in Line 10 of Algorithm 1 to obtain

$$\frac{\gamma + \frac{1}{2}\eta\mu}{p\eta}\mathbb{E}\left[\|w^{k+1} - z^*\|^2\right] = \frac{\gamma + \frac{1}{2}\eta\mu}{p\eta}\mathbb{E}\left[\mathbb{E}_{w^{k+1}}\left[\|w^{k+1} - z^*\|^2\right]\right]$$

$$= \frac{\gamma + \frac{1}{2}\eta\mu}{\eta}\mathbb{E}\left[\|z^{k+1} - z^*\|^2\right] + \frac{(\gamma + \frac{1}{2}\eta\mu)(1 - p)}{p\eta}\mathbb{E}\left[\|w^k - z^*\|^2\right],$$

and

$$\frac{\gamma}{\eta} + \frac{(\gamma + \frac{1}{2}\eta\mu)(1 - p)}{p\eta} = \left(1 - p + \frac{p\gamma}{(\gamma + \frac{1}{2}\eta\mu)}\right)\frac{(\gamma + \frac{1}{2}\eta\mu)}{p\eta} = \left(1 - \frac{p\eta\mu}{2\gamma + \eta\mu}\right)\frac{(\gamma + \frac{1}{2}\eta\mu)}{p\eta}.$$

Combining all above results, we achieve

$$\left(\frac{1 - \gamma}{\eta} + \frac{\mu}{2}\right)\mathbb{E}\left[\|z^{k+1} - z^*\|^2\right] + 2\mathbb{E}\left[\langle F(z^k) - F_1(z^k) - F(z^{k+1}) + F_1(z^{k+1}), z^{k+1} - z^*\rangle\right]$$

$$+ \frac{1}{64\eta}\mathbb{E}\left[\|z^{k+1} - z^k\|^2\right] + \frac{\gamma}{4\eta}\mathbb{E}\left[\|w^k - z^{k+1}\|^2\right] + \frac{\gamma + \frac{1}{2}\eta\mu}{p\eta}\mathbb{E}\left[\|w^{k+1} - z^*\|^2\right]$$

$$\leq \frac{1 - \gamma}{\eta}\mathbb{E}\left[\|z^k - z^*\|^2\right] + 2\alpha\mathbb{E}\left[\langle F(z^{k-1}) - F_1(z^{k-1}) - F(z^k) + F_1(z^k), z^k - z^*\rangle\right]$$

$$+ \frac{\alpha}{64\eta}\mathbb{E}\left[\|z^k - z^{k-1}\|^2\right] + \frac{4\eta\delta^2}{b}\mathbb{E}\left[\|w^{k-1} - z^k\|^2\right] - \frac{1}{16\eta}\mathbb{E}\left[\|z^k - \hat{u}^k\|^2\right]$$

$$+ \left(-\frac{7}{8\eta} + \frac{2\delta^2}{\mu} + 64\eta\delta^2 + 3\mu\right)\mathbb{E}\left[\|\hat{u}^k - u^k\|^2\right] + \frac{2}{\eta}\mathbb{E}\left[\|\hat{u}^k - u^k\|\|z^{k+1} - z^*\|\right]$$

$$+ \left(1 - \frac{p\eta\mu}{2\gamma + \eta\mu}\right)\frac{(\gamma + \frac{1}{2}\eta\mu)}{p\eta}\mathbb{E}\left[\|w^k - z^*\|^2\right] - \frac{\gamma}{2\eta}\mathbb{E}\left[\|w^k - \hat{u}^k\|^2\right].$$

$\square$

**Lemma 8.** *Under setting of Lemma 1, we additionally assume $\mathbb{E}[\Phi^k] \le \Phi^0$ holds, then we have*

$$\mathbb{E}[\Phi^{k+1}] \le \max\left\{1 - \frac{\eta\mu}{6(1-\gamma)}, 1 - \frac{p\eta\mu}{2\gamma + \eta\mu}\right\}\mathbb{E}[\Phi^k] - \frac{1}{16\eta}\mathbb{E}\left[\|z^k - \hat{u}^k\|^2\right] - \frac{\gamma}{2\eta}\mathbb{E}\left[\|w^k - \hat{u}^k\|^2\right].$$

*Proof.* Recall that the definition of our Lyapunov function is

$$\Phi^k = \left(\frac{1-\gamma}{\eta} + \frac{\mu}{2}\right)\|z^k - z^*\|^2 + 2\langle F(z^{k-1}) - F_1(z^{k-1}) - F(z^k) + F_1(z^k), z^k - z^*\rangle$$

$$+ \frac{1}{64\eta}\|z^k - z^{k-1}\|^2 + \frac{\gamma}{4\eta}\|w^{k-1} - z^k\|^2 + \frac{(2\gamma + \eta\mu)}{2p\eta}\|w^k - z^*\|^2. \tag{22}$$

Recall that we take constant $c$ by equation (14), then the condition

$$\varepsilon_k \le c^{-1}\min\left\{\|\hat{u}^k - z^k\|, \|\hat{u}^k - z^k\|^2\right\}$$

guarantees

$$\mathbb{E}\left[\|\hat{u}^k - u^k\|\right] \le \tilde{\zeta}^k \min\left\{\mathbb{E}\left[\|\hat{u}^k - z^k\|\right], \mathbb{E}\left[\|\hat{u}^k - z^k\|^2\right]\right\},$$

where

$$\begin{aligned}
\tilde{\zeta}^k &= \frac{1}{32\eta}\frac{1}{\frac{9}{8\eta} + \frac{2\delta^2}{\mu} + 64\eta\delta^2 + 3\mu + \frac{2}{\eta} + 2\sqrt{\frac{2\mathbb{E}[\Phi^k]}{\eta}}}\\
&= \frac{1}{100 + \frac{64\eta\delta^2}{\mu} + 2048\eta^2\delta^2 + 96\eta\mu + 64\sqrt{2\eta\mathbb{E}[\Phi^k]}} \tag{23}\\
&\ge \frac{1}{100 + \frac{64\eta\delta^2}{\mu} + 2048\eta^2\delta^2 + 96\eta\mu + 64\sqrt{2\eta\Phi^0}} = \frac{1}{c}.
\end{aligned}$$

The inequality (23) is based on the assumption $\mathbb{E}[\Phi^k] \le \Phi^0$.

Note that we have $\|z^{k+1} - z^*\| \le \|u^k - \hat{u}^k\| + \|\hat{u}^k - z^k\| + \|z^k - z^*\|$, then

$$\begin{aligned}
&\left(-\frac{7}{8\eta} + \frac{2\delta^2}{\mu} + 64\eta\delta^2 + 3\mu\right)\mathbb{E}\left[\|\hat{u}^k - u^k\|^2\right] + \frac{2}{\eta}\mathbb{E}\left[\|\hat{u}^k - u^k\|\|z^{k+1} - z^*\|\right]\\
&\le \left(\frac{9}{8\eta} + \frac{2\delta^2}{\mu} + 64\eta\delta^2 + 3\mu\right)\mathbb{E}\left[\|\hat{u}^k - u^k\|^2\right] + \frac{2}{\eta}\mathbb{E}\left[\|\hat{u}^k - u^k\|\|\hat{u}^k - z^k\|\right]\\
&\quad + \frac{2}{\eta}\mathbb{E}\left[\|\hat{u}^k - u^k\|\|z^k - z^*\|\right]\\
&\le \left(\frac{9}{8\eta} + \frac{2\delta^2}{\mu} + 64\eta\delta^2 + 3\mu\right)\mathbb{E}\left[\|\hat{u}^k - u^k\|^2\right] + \frac{2}{\eta}\mathbb{E}\left[\|\hat{u}^k - u^k\|\|\hat{u}^k - z^k\|\right]\\
&\quad + 2\sqrt{\frac{2\mathbb{E}[\Phi^k]}{\eta}}\mathbb{E}\left[\|\hat{u}^k - u^k\|\right]\\
&\le \frac{1}{32\eta}\mathbb{E}\left[\|\hat{u}^k - z^k\|^2\right].
\end{aligned}$$

According to Lemma 7, we have

$$\begin{aligned}
&\left(\frac{1-\gamma}{\eta} + \frac{\mu}{2}\right)\mathbb{E}\left[\|z^{k+1} - z^*\|^2\right] + 2\mathbb{E}\left[\langle F(z^k) - F_1(z^k) - F(z^{k+1}) + F_1(z^{k+1}), z^{k+1} - z^*\rangle\right]\\
&\quad + \frac{1}{64\eta}\mathbb{E}\left[\|z^{k+1} - z^k\|^2\right] + \frac{\gamma}{4\eta}\mathbb{E}\left[\|w^k - z^{k+1}\|^2\right] + \frac{\gamma + \frac{1}{2}\eta\mu}{p\eta}\mathbb{E}\left[\|w^{k+1} - z^*\|^2\right]\\
&\le \frac{1-\gamma}{\eta}\mathbb{E}\left[\|z^k - z^*\|^2\right] + 2\alpha\mathbb{E}\left[\langle F(z^{k-1}) - F_1(z^{k-1}) - F(z^k) + F_1(z^k), z^k - z^*\rangle\right]\\
&\quad + \alpha\frac{1}{64\eta}\mathbb{E}\left[\|z^k - z^{k-1}\|^2\right] + \frac{4\eta\delta^2}{b}\mathbb{E}\left[\|w^{k-1} - z^k\|^2\right] + \frac{1}{32\eta}\mathbb{E}\left[\|z^k - \hat{u}^k\|^2\right]\\
&\quad - \frac{1}{16\eta}\mathbb{E}\left[\|z^k - \hat{u}^k\|^2\right] - \frac{\gamma}{2\eta}\mathbb{E}\left[\|w^k - \hat{u}^k\|^2\right] + \left(1 - \frac{p\eta\mu}{2\gamma + \eta\mu}\right)\frac{(\gamma + \frac{1}{2}\eta\mu)}{p\eta}\mathbb{E}\left[\|w^k - z^*\|^2\right].
\end{aligned}$$

From the fact that $\eta\mu \leq 1$, we have

$$\frac{1-\gamma}{\eta} \leq \left(1 - \frac{\eta\mu}{6(1-\gamma)}\right)\left(\frac{1-\gamma}{\eta} + \frac{\mu}{2}\right),$$

and according to the fact that $4\eta\delta^2/b \leq \alpha\gamma/(4\eta)$, we obtain

$$\left(\frac{1-\gamma}{\eta} + \frac{\mu}{2}\right)\mathbb{E}\left[\|z^{k+1} - z^*\|^2\right] + 2\mathbb{E}\left[\langle F(z^k) - F_1(z^k) - F(z^{k+1}) + F_1(z^{k+1}), z^{k+1} - z^*\rangle\right]$$

$$+ \frac{1}{64\eta}\mathbb{E}\left[\|z^{k+1} - z^k\|^2\right] + \frac{\gamma}{4\eta}\mathbb{E}\left[\|w^k - z^{k+1}\|^2\right] + \frac{\gamma + \frac{1}{2}\eta\mu}{p\eta}\mathbb{E}\left[\|w^{k+1} - z^*\|^2\right]$$

$$\leq \left(1 - \frac{\eta\mu}{6(1-\gamma)}\right)\left(\frac{1-\gamma}{\eta} + \frac{\mu}{2}\right)\mathbb{E}\left[\|z^k - z^*\|^2\right] + \left(1 - \frac{p\eta\mu}{2\gamma + \eta\mu}\right)\frac{(\gamma + \frac{1}{2}\eta\mu)}{p\eta}\mathbb{E}\left[\|w^k - z^*\|^2\right]$$

$$+ 2\alpha\mathbb{E}\left[\langle F(z^{k-1}) - F_1(z^{k-1}) - F(z^k) + F_1(z^k), z^k - z^*\rangle\right] + \alpha\frac{1}{64\eta}\mathbb{E}\left[\|z^k - z^{k-1}\|^2\right]$$

$$+ \alpha\frac{\gamma}{2\eta}\mathbb{E}\left[\|w^{k-1} - z^k\|^2\right] - \frac{1}{16\eta}\mathbb{E}\left[\|z^k - \hat{u}^k\|^2\right] - \frac{\gamma}{2\eta}\mathbb{E}\left[\|w^k - \hat{u}^k\|^2\right].$$

The definition (22) and the setting $\alpha = \max\{1 - \eta\mu/(6(1-\gamma)), 1 - p\eta\mu/(2\gamma + \eta\mu)\}$ implies

$$\mathbb{E}[\Phi^{k+1}] \leq \alpha\mathbb{E}[\Phi^k] - \frac{1}{16\eta}\mathbb{E}\left[\|z^k - \hat{u}^k\|^2\right] - \frac{\gamma}{2\eta}\mathbb{E}\left[\|w^k - \hat{u}^k\|^2\right]. \tag{24}$$

$\square$

Then we provide the proof of Lemma 1.

*Proof.* We firstly use the induction to prove

$$\mathbb{E}[\Phi^k] \leq \Phi^0$$

holds for all $k \in \mathbb{N}$.

Note that it holds for $k = 0$. Assume we have $\mathbb{E}[\Phi^k] \leq \Phi^0$ holds, then Lemma 8 means it holds

$$\mathbb{E}[\Phi^{k+1}] \leq \max\left\{1 - \frac{\eta\mu}{6(1-\gamma)}, 1 - \frac{p\eta\mu}{2\gamma + \eta\mu}\right\}\mathbb{E}[\Phi^k] \leq \mathbb{E}[\Phi^k] \leq \Phi^0,$$

which finish the induction.

The result of above induction implies the condition of Lemma 8 always holds. Therefore, we can apply Lemma 8 to achieve equation (24), which finishes the proof of Lemma 1. $\square$

## C.2 Proof of Theorem 1

We firstly introduce the following quantities for our analysis

$$e_{11}(z, k) := \frac{2\eta}{b}\sum_{j \in \mathcal{S}^k}\langle F(z^k) - F_j(z^k) - F(w^{k-1}) + F_j(w^{k-1}), \hat{u}^k - z\rangle,$$

$$e_{12}(z, k) := \frac{2\eta\alpha}{b}\sum_{j \in \mathcal{S}^k}\langle F(z^k) - F_j(z^k) - F(z^{k-1}) + F_j(z^{k-1}), \hat{u}^k - z\rangle, \tag{25}$$

$$e_2(z, k) := \|w^{k+1} - z\|^2 - p\|z^k - z\|^2 - (1-p)\|w^k - z\|^2,$$

$$\Psi^k(z) := (1-\gamma)\|z^{k+1} - z\|^2 + \frac{\gamma}{p}\|w^k - z\|^2 + \frac{1}{16}\|z^k - z^{k-1}\|^2.$$

Specifically, we take the constant

$$\zeta := \min\left\{\frac{\eta^2\varepsilon^2}{16(9\eta LD + 3\eta\max_{i \in [n]}\|F_i(z^0)\| + D)^2}, \frac{\eta\varepsilon}{4(12\eta^2\delta^2 + 1)}\right\} \tag{26}$$

and

$$\hat{c} := 100 + 2048\eta^2\delta^2 + 64\sqrt{2\eta\Phi^0} \le 102 + 16\sqrt{\Phi^0/\delta} \tag{27}$$

for the statement Theorem 1. We then provide several lemmas that will be used in the proof of Theorem 1.

**Lemma 9.** *Under the setting of Theorem 1, we have*

$$-2\langle \mathbb{E}_k[\delta^k] + F_1(\hat{u}^k), \hat{u}^k - z\rangle$$

$$\le 4LD\|u^k - \hat{u}^k\| + 2\langle F(u^k), z - u^k\rangle + (8LD + 6D_F)\|\hat{u}^k - u^k\| + \frac{1}{16\eta}\|z^k - z^{k+1}\|^2$$

$$+ 16\eta\delta^2\|\hat{u}^k - u^k\|^2 - 2\langle F(z^k) - F_1(z^k) - F(z^{k+1}) + F_1(z^{k+1}), z^{k+1} - z\rangle$$

$$+ \frac{1}{2\eta}\|z^k - \hat{u}^k\|^2 + 2\eta\delta^2\alpha^2\|z^k - z^{k-1}\|^2$$

$$- 2\alpha\langle F(z^k) - F_1(z^k) - F(z^{k-1}) + F_1(z^{k-1}), z^k - z\rangle,$$

*where* $D_F := \max_{i \in [n]} \sup_{z \in \mathcal{Z}} \|F_i(z)\|$.

*Proof.* Note that the sequence $\{\|F_i(z)\|\}_{i=1}^n$ is bounded on $z \in \mathcal{Z}$, since we have

$$D_F = \max_{i \in [n]} \sup_{z \in \mathcal{Z}} \|F_i(z)\| \le \max_{i \in [n]} \sup_{z \in \mathcal{Z}} (\|F_i(z) - F_i(z^0)\| + \|F_i(z^0)\|) \le LD + \max_{i \in [n]} \|F_i(z^0)\|. \tag{28}$$

The Lipschitz continuity of $F(\cdot)$ impies

$$\|F_1(\hat{u}^k)\| - D_F \le \|F_1(\hat{u}^k)\| - \|F_1(z)\| \le \|F_1(\hat{u}^k) - F_1(z)\| \le LD,$$

then we have

$$-2\langle \mathbb{E}_k[\delta^k] + F_1(\hat{u}^k), \hat{u}^k - z\rangle$$

$$= -2\langle F(z^k) - F_1(z^k) + \alpha\left(F(z^k) - F_1(z^k) - F(z^{k-1}) + F_1(z^{k-1})\right) + F_1(\hat{u}^k), \hat{u}^k - z\rangle$$

$$= -2\langle F(u^k) - F_1(u^k) - F(\hat{u}^k) + F_1(\hat{u}^k), \hat{u}^k - z\rangle$$

$$\quad - 2\langle F(\hat{u}^k), \hat{u}^k - z\rangle$$

$$\quad - 2\langle F(z^k) - F_1(z^k) - F(z^{k+1}) + F_1(z^{k+1}), \hat{u}^k - z^{k+1}\rangle$$

$$\quad - 2\langle F(z^k) - F_1(z^k) - F(z^{k+1}) + F_1(z^{k+1}), z^{k+1} - z\rangle$$

$$\quad - 2\alpha\langle F(z^k) - F_1(z^k) - F(z^{k-1}) + F_1(z^{k-1}), \hat{u}^k - z^k\rangle \tag{29}$$

$$\quad - 2\alpha\langle F(z^k) - F_1(z^k) - F(z^{k-1}) + F_1(z^{k-1}), z^k - z\rangle$$

$$\le 4LD\|u^k - \hat{u}^k\| + 2\langle F(u^k), z - u^k\rangle + (8LD + 6D_F)\|\hat{u}^k - u^k\| + \frac{1}{16\eta}\|z^k - z^{k+1}\|^2$$

$$+ 16\eta\delta^2\|\hat{u}^k - u^k\|^2 - 2\langle F(z^k) - F_1(z^k) - F(z^{k+1}) + F_1(z^{k+1}), z^{k+1} - z\rangle$$

$$+ \frac{1}{2\eta}\|z^k - \hat{u}^k\|^2 + 2\eta\delta^2\alpha^2\|z^k - z^{k-1}\|^2$$

$$- 2\alpha\langle F(z^k) - F_1(z^k) - F(z^{k-1}) + F_1(z^{k-1}), z^k - z\rangle.$$

$\square$

**Lemma 10.** *Under the setting of Theorem 1, the quantities defined in equation (25) hold*

$$\max_{z \in \mathcal{Z}} \Psi^0(z) + \mathbb{E}\left[\max_{z \in \mathcal{Z}} \sum_{k=0}^{K-1} e_{11}(z, k) + e_{12}(z, k) + \frac{\gamma}{p}e_2(z, k)\right]$$

$$\le \max_{z \in \mathcal{Z}} 4\|z^0 - z\|^2 + \frac{4\eta^2\delta^2}{b}\sum_{k=0}^{K-1}\mathbb{E}\left[\|z^k - \hat{u}^k\|^2\right] + \left(2p + \frac{4\eta^2\delta^2}{b}\right)\sum_{k=0}^{K-1}\mathbb{E}\left[\|w^k - \hat{u}^k\|^2\right]$$

$$+ \left(2p + \frac{8\eta^2\delta^2}{b}\right)\sum_{k=0}^{K-1}\mathbb{E}\left[\|\hat{u}^k - u^k\|^2\right].$$

*Proof.* Applying Lemma 4 with $x^0 = z^0$, $\mathcal{F}_0 = \sigma(\mathcal{S}^0)$, $\mathcal{F}_k = \sigma(\mathcal{S}^0, \ldots, \mathcal{S}^{k-1}, w^k)$ for $k \geq 1$, and $r^{k+1} = \frac{2\eta}{b}\sum_{j \in \mathcal{S}^k} F_j(z^k) - F(z^k) - F_j(w^{k-1}) + F(w^{k-1})$ and using $\mathbb{E}_k[r^{k+1}] = 0$, we have

$$\mathbb{E}\left[\max_{z \in \mathcal{Z}}\sum_{k=0}^{K-1} e_{11}(z,k)\right] = \mathbb{E}\left[\max_{z \in \mathcal{Z}}\sum_{k=0}^{K-1}\langle r^{k+1}, z\rangle\right] \leq \max_{z \in \mathcal{Z}}\frac{1}{2}\|z^0 - z\|^2 + \frac{1}{2}\sum_{k=0}^{K-1}\mathbb{E}\left[\|r^{k+1}\|^2\right]$$

$$\leq \max_{z \in \mathcal{Z}}\frac{1}{2}\|z^0 - z\|^2 + \frac{2\eta^2\delta^2}{b}\sum_{k=0}^{K-1}\mathbb{E}\left[\|z^k - w^{k-1}\|^2\right].$$

Similarly we can obtain

$$\mathbb{E}\left[\max_{z \in \mathcal{Z}}\sum_{k=0}^{K-1} e_{12}(z,k)\right] \leq \max_{z \in \mathcal{Z}}\frac{1}{2}\|z^0 - z\|^2 + \frac{2\eta^2\alpha^2\delta^2}{b}\sum_{k=0}^{K-1}\mathbb{E}\left[\|z^k - z^{k-1}\|^2\right].$$

Applying Lemma 4 with $x^0 = z^0$, $\mathcal{F}_0 = \sigma(\mathcal{S}^0)$, $\mathcal{F}_k = \sigma(\mathcal{S}^0, \ldots, \mathcal{S}^{k-1}, w^k)$ for $k \geq 1$, and $r^{k+1} = pz^{k+1} + (1-p)w^k - w^{k+1}$ and using the fact that $\mathbb{E}[\|w^{k+1}\|^2 - p\|z^{k+1}\|^2 - (1-p)\|w^k\|^2] = 0$ and $\mathbb{E}_k[r^{k+1}] = 0$, we have

$$\mathbb{E}\left[\max_{z \in \mathcal{Z}}\sum_{k=0}^{K-1} e_2(z,k)\right] = 2\mathbb{E}\left[\max_{z \in \mathcal{Z}}\sum_{k=0}^{K-1}\langle r^{k+1}, z\rangle\right] \leq \max_{z \in \mathcal{Z}}\|z^0 - z\|^2 + \sum_{k=0}^{K-1}\mathbb{E}\left[\|r^{k+1}\|^2\right]$$

$$\leq \max_{z \in \mathcal{Z}}\|z^0 - z\|^2 + p(1-p)\sum_{k=0}^{K-1}\mathbb{E}\left[\|z^{k+1} - w^k\|^2\right],$$

where we use

$$\mathbb{E}\left[\|r^{k+1}\|^2\right] = \mathbb{E}\left[\mathbb{E}_{k+1/2}\|\mathbb{E}_{k+1/2}[w^{k+1}] - w^{k+1}\|^2\right]$$

$$= \mathbb{E}\left[\mathbb{E}_{k+1/2}[\|w^{k+1}\|^2] - \|\mathbb{E}_{k+1/2}[w^{k+1}]\|^2\right]$$

$$= \mathbb{E}\left[p\|z^{k+1}\|^2 + (1-p)\|w^k\|^2 - \|pz^{k+1} + (1-p)w^k\|^2\right]$$

$$= p(1-p)\mathbb{E}\left[\|z^{k+1} - w^k\|^2\right].$$

Note that $z^k = u^{k-1}$, then we have

$$\max_{z \in \mathcal{Z}}\Psi^0(z) + \mathbb{E}\left[\max_{z \in \mathcal{Z}}\sum_{k=0}^{K-1} e_{11}(z,k) + e_{12}(z,k) + \frac{\gamma}{p}e_2(z,k)\right]$$

$$\leq 4\max_{z \in \mathcal{Z}}\|z^0 - z\|^2 + \frac{2\eta^2\delta^2}{b}\sum_{k=0}^{K-1}\mathbb{E}\left[\|z^k - w^{k-1}\|^2\right] + \frac{2\eta^2\alpha^2\delta^2}{b}\sum_{k=0}^{K-1}\mathbb{E}\left[\|z^k - z^{k-1}\|^2\right]$$

$$+ p(1-p)\sum_{k=0}^{K-1}\mathbb{E}\left[\|z^{k+1} - w^k\|^2\right]$$

$$\leq 4\max_{z \in \mathcal{Z}}\|z^0 - z\|^2 + \frac{4\eta^2\delta^2}{b}\sum_{k=0}^{K-1}\mathbb{E}\left[\|\hat{u}^{k-1} - w^{k-1}\|^2\right] + \frac{4\eta^2\alpha^2\delta^2}{b}\sum_{k=0}^{K-1}\mathbb{E}\left[\|\hat{u}^{k-1} - z^{k-1}\|^2\right]$$

$$+ 2p\sum_{k=0}^{K-1}\mathbb{E}\left[\|\hat{u}^k - w^k\|^2\right] + \left(\frac{4\eta^2\delta^2}{b} + \frac{4\eta^2\alpha^2\delta^2}{b} + 2p\right)\sum_{k=0}^{K-1}\mathbb{E}\left[\|\hat{u}^k - u^k\|^2\right]$$

$$\leq 4\max_{z \in \mathcal{Z}}\|z^0 - z\|^2 + \frac{4\eta^2\delta^2}{b}\sum_{k=0}^{K-1}\mathbb{E}\left[\|z^k - \hat{u}^k\|^2\right] + \left(2p + \frac{4\eta^2\delta^2}{b}\right)\sum_{k=0}^{K-1}\mathbb{E}\left[\|w^k - \hat{u}^k\|^2\right]$$

$$+ \left(2p + \frac{8\eta^2\delta^2}{b}\right)\sum_{k=0}^{K-1}\mathbb{E}\left[\|\hat{u}^k - u^k\|^2\right],$$

where we use Young's inequality and $\alpha \leq 1$. $\qquad\square$

Now we provide the proof of Theorem 1.

*Proof.* The optimality of $\hat{u}^k$ implies for all $z \in \mathcal{Z}$, we have

$$\langle \eta F_1(\hat{u}^k) + \hat{u}^k - v^k, z - \hat{u}^k \rangle \geq 0. \tag{30}$$

Combine equation (30) with the update rule in Line 7 of Algorithm 1 , we achieve

$$-\frac{1}{\eta}\langle \bar{z}^k - \hat{u}^k - \eta\delta^k, \hat{u}^k - z \rangle \leq -\langle F_1(\hat{u}^k), \hat{u}^k - z \rangle. \tag{31}$$

Then we have

$$
\begin{aligned}
\frac{1}{\eta}\|\hat{u}^k - z\|^2 &= \frac{1}{\eta}\|z^k - z\|^2 + \frac{2}{\eta}\langle \hat{u}^k - z^k, \hat{u}^k - z \rangle - \frac{1}{\eta}\|\hat{u}^k - z^k\|^2 \\
&= \frac{1}{\eta}\|z^k - z\|^2 + \frac{2\gamma}{\eta}\langle w^k - z^k, \hat{u}^k - z \rangle - 2\langle \delta^k, \hat{u}^k - z \rangle \\
&\quad - \frac{1}{\eta}\|\hat{u}^k - z^k\|^2 - \frac{2}{\eta}\langle \bar{z}^k - \hat{u}^k - \eta\delta^k, \hat{u}^k - z \rangle \\
&\leq \frac{1}{\eta}\|z^k - z\|^2 + \frac{2\gamma}{\eta}\langle w^k - z^k, \hat{u}^k - z \rangle - 2\langle \delta^k, \hat{u}^k - z \rangle - \frac{1}{\eta}\|\hat{u}^k - z^k\|^2 \\
&\quad - 2\langle F_1(\hat{u}^k), \hat{u}^k - z \rangle \\
&= \frac{1}{\eta}\|z^k - z\|^2 + \frac{\gamma}{\eta}\|w^k - z\|^2 - \frac{\gamma}{\eta}\|w^k - \hat{u}^k\|^2 - \frac{\gamma}{\eta}\|z^k - z\|^2 - \frac{1-\gamma}{\eta}\|\hat{u}^k - z^k\|^2 \\
&\quad - 2\langle \delta^k + F_1(\hat{u}^k), \hat{u}^k - z \rangle \\
&= \frac{1}{\eta}\|z^k - z\|^2 + \frac{\gamma}{\eta}\|w^k - z\|^2 - \frac{\gamma}{\eta}\|w^k - \hat{u}^k\|^2 - \frac{\gamma}{\eta}\|z^k - z\|^2 - \frac{1-\gamma}{\eta}\|\hat{u}^k - z^k\|^2 \\
&\quad - 2\langle \mathbb{E}\delta^k + F_1(\hat{u}^k), \hat{u}^k - z \rangle + \frac{1}{\eta}e_{11}(z, k) + \frac{1}{\eta}e_{12}(z, k),
\end{aligned}
\tag{32}
$$

Combining the results of equation (32) with Lemma 9, we have

$$
\begin{aligned}
\frac{1}{\eta}\|\hat{u}^k - z\|^2 &\leq \frac{1}{\eta}\|z^k - z\|^2 + \frac{\gamma}{\eta}\|w^k - z\|^2 - \frac{\gamma}{\eta}\|w^k - \hat{u}^k\|^2 - \frac{\gamma}{\eta}\|z^k - z\|^2 \\
&\quad - \frac{1-\gamma}{\eta}\|\hat{u}^k - z^k\|^2 - 2\langle \mathbb{E}\delta^k + F_1(\hat{u}^k), \hat{u}^k - z \rangle + \frac{1}{\eta}e_{11}(z, k) + \frac{1}{\eta}e_{12}(z, k) \\
&\leq \frac{1}{\eta}\|z^k - z\|^2 + \frac{\gamma}{\eta}\|w^k - z\|^2 - \frac{\gamma}{\eta}\|w^k - \hat{u}^k\|^2 - \frac{\gamma}{\eta}\|z^k - z\|^2 - \frac{1}{4\eta}\|\hat{u}^k - z^k\|^2 \\
&\quad + 2\langle F(u^k), z - u^k \rangle + (12LD + 6D_F)\|\hat{u}^k - u^k\| + \frac{1}{16\eta}\|z^k - z^{k+1}\|^2 \\
&\quad + 16\eta\delta^2\|\hat{u}^k - u^k\|^2 - 2\langle F(z^k) - F_1(z^k) - F(z^{k+1}) + F_1(z^{k+1}), z^{k+1} - z \rangle \\
&\quad + 2\eta\delta^2\alpha^2\|z^k - z^{k-1}\|^2 - 2\alpha\langle F(z^k) - F_1(z^k) - F(z^{k-1}) + F_1(z^{k-1}), z^k - z \rangle \\
&\quad + \frac{1}{\eta}e_{11}(z, k) + \frac{1}{\eta}e_{12}(z, k),
\end{aligned}
\tag{33}
$$

where we use the fact $\gamma \leq 1/4$.

From the fact that $\|a + b\|^2 \geq \frac{1}{2}\|a\|^2 - \|b\|^2$ and $\frac{3}{2}\|a + b\|^2 \geq \|a\|^2 - 3\|b\|^2$ , we have

$$
\begin{aligned}
-\frac{1}{4\eta}\|\hat{u}^k - z^k\|^2 &\leq -\frac{1}{8\eta}\|z^{k+1} - z^k\|^2 + \frac{1}{4\eta}\|\hat{u}^k - u^k\|^2, \\
-\frac{\gamma}{\eta}\|w^k - \hat{u}^k\|^2 &\leq -\frac{\gamma}{2\eta}\|w^k - z^{k+1}\|^2 + \frac{\gamma}{\eta}\|\hat{u}^k - u^k\|^2.
\end{aligned}
\tag{34}
$$

Plugging equation (34) into equation (33), we achieve

$$
\begin{aligned}
\frac{1}{\eta}\|\hat{u}^k - z\|^2 \leq{} & \frac{1}{\eta}\|z^k - z\|^2 + \frac{\gamma}{\eta}\|w^k - z\|^2 - \frac{\gamma}{\eta}\|z^k - z\|^2 - \frac{1}{8\eta}\|z^{k+1} - z^k\|^2 \\
& + \frac{1}{4\eta}\|\hat{u}^k - u^k\|^2 + 2\langle F(u^k), z - u^k\rangle + (12LD + 6D_F)\|\hat{u}^k - u^k\| \\
& + \frac{1}{16\eta}\|z^k - z^{k+1}\|^2 - 2\langle F(z^k) - F_1(z^k) - F(z^{k+1}) + F_1(z^{k+1}), z^{k+1} - z\rangle \\
& + 16\eta\delta^2\|\hat{u}^k - u^k\|^2 - 2\alpha\langle F(z^k) - F_1(z^k) - F(z^{k-1}) + F_1(z^{k-1}), z^k - z\rangle \\
& + 2\eta\delta^2\alpha^2\|z^k - z^{k-1}\|^2 - \frac{\gamma}{2\eta}\|w^k - z^{k+1}\|^2 + \frac{\gamma}{\eta}\|\hat{u}^k - u^k\|^2 \\
& + \frac{1}{\eta}e_{11}(z, k) + \frac{1}{\eta}e_{12}(z, k) \\
\leq{} & \frac{1}{\eta}\|z^k - z\|^2 + \frac{\gamma}{\eta}\|w^k - z\|^2 - \frac{\gamma}{\eta}\|z^k - z\|^2 - \frac{1}{16\eta}\|z^{k+1} - z^k\|^2 \\
& - \frac{\gamma}{2\eta}\|w^k - z^{k+1}\|^2 + 2\langle F(u^k), z - u^k\rangle + (12LD + 6D_F)\|\hat{u}^k - u^k\| \\
& - 2\langle F(z^k) - F_1(z^k) - F(z^{k+1}) + F_1(z^{k+1}), z^{k+1} - z\rangle + 2\eta\delta^2\alpha^2\|z^k - z^{k-1}\|^2 \\
& - 2\alpha\langle F(z^k) - F_1(z^k) - F(z^{k-1}) + F_1(z^{k-1}), z^k - z\rangle \\
& + \left(16\eta\delta^2 + \frac{1}{2\eta}\right)\|\hat{u}^k - u^k\|^2 + \frac{1}{\eta}e_{11}(z, k) + \frac{1}{\eta}e_{12}(z, k).
\end{aligned}
$$

Note that the fact

$$
\frac{1}{\eta}\|\hat{u}^k - z\|^2 = \frac{1}{\eta}\|z^{k+1} - z\|^2 + \frac{1}{\eta}\|\hat{u}^k - u^k\|^2 - \frac{2}{\eta}\|\hat{u}^k - u^k\|\|z^{k+1} - z\|,
$$

then we have

$$
\begin{aligned}
2\langle F(u^k), u^k - z\rangle \leq{} & -\frac{1}{\eta}\|z^{k+1} - z\|^2 - \frac{1}{\eta}\|\hat{u}^k - u^k\|^2 + \frac{2}{\eta}\|\hat{u}^k - u^k\|\|z^{k+1} - z\| \\
& + \frac{1}{\eta}\|z^k - z\|^2 + \frac{\gamma}{\eta}\|w^k - z\|^2 - \frac{\gamma}{\eta}\|z^k - z\|^2 - \frac{1}{16\eta}\|z^{k+1} - z^k\|^2 \\
& - \frac{\gamma}{2\eta}\|w^k - z^{k+1}\|^2 + (12LD + 6D_F)\|\hat{u}^k - u^k\| + 2\eta\delta^2\alpha^2\|z^k - z^{k-1}\|^2 \\
& - 2\langle F(z^k) - F_1(z^k) - F(z^{k+1}) + F_1(z^{k+1}), z^{k+1} - z\rangle \\
& - 2\alpha\langle F(z^k) - F_1(z^k) - F(z^{k-1}) + F_1(z^{k-1}), z^k - z\rangle \\
& + \left(16\eta\delta^2 + \frac{1}{2\eta}\right)\|\hat{u}^k - u^k\|^2 + \frac{1}{\eta}e_{11}(z, k) + \frac{1}{\eta}e_{12}(z, k) \\
\leq{} & -\frac{1}{\eta}\|z^{k+1} - z\|^2 - \frac{1}{16\eta}\|z^{k+1} - z^k\|^2 - \frac{\gamma}{2\eta}\|z^{k+1} - w^k\|^2 \\
& - 2\langle F(z^k) - F_1(z^k) - F(z^{k+1}) + F_1(z^{k+1}), z^{k+1} - z\rangle \\
& + \frac{1}{\eta}\|z^k - z\|^2 + \frac{\gamma}{\eta}\|w^k - z\|^2 - \frac{\gamma}{\eta}\|z^k - z\|^2 + 2\eta\delta^2\alpha^2\|z^k - z^{k-1}\|^2 \\
& + 2\alpha\langle F(z^{k-1}) - F_1(z^{k-1}) - F(z^k) + F_1(z^k), z^k - z\rangle \\
& + \left(12LD + 6D_F + \frac{2D}{\eta}\right)\|\hat{u}^k - u^k\| + 16\eta\delta^2\|\hat{u}^k - u^k\|^2 \\
& + \frac{1}{\eta}e_{11}(z, k) + \frac{1}{\eta}e_{12}(z, k) \\
\leq{} & -\frac{1 - \gamma}{\eta}\|z^{k+1} - z\|^2 - \frac{\gamma}{\eta p}\|w^{k+1} - z\|^2 - \frac{1}{16\eta}\|z^{k+1} - z^k\|^2 \\
& - 2\langle F(z^k) - F_1(z^k) - F(z^{k+1}) + F_1(z^{k+1}), z^{k+1} - z\rangle
\end{aligned}
$$

$$
+ \frac{1-\gamma}{\eta} \|z^k - z\|^2 + \frac{\gamma}{\eta p} \|w^k - z\|^2 + 2\eta\delta^2\alpha^2 \|z^k - z^{k-1}\|^2
$$
$$
+ 2\alpha \langle F(z^{k-1}) - F_1(z^{k-1}) - F(z^k) + F_1(z^k), z^k - z \rangle
$$
$$
+ \left( 12L\Omega + 6D_F + \frac{2D}{\eta} \right) \|\hat{u}^k - u^k\| + 16\eta\delta^2 \|\hat{u}^k - u^k\|^2
$$
$$
+ \frac{1}{\eta} \left( e_{11}(z,k) + e_{12}(z,k) + \frac{\gamma}{p} e_2(z,k) \right) - \frac{\gamma}{2\eta} \|z^{k+1} - w^k\|^2.
$$

The parameters settings implies $2\eta\delta^2\alpha^2 \le 1/(16\eta)$, then we have

$$
2\eta K \mathbb{E} \left[ \max_{z \in \mathcal{Z}} \frac{1}{K} \sum_{k=0}^{K-1} \langle F(u^k), u^k - z \rangle \right]
$$
$$
\le \max_{z \in \mathcal{Z}} \Psi^0(z) + \mathbb{E} \left[ \max_{z \in \mathcal{Z}} \sum_{k=0}^{K-1} e_{11}(z,k) + e_{12}(z,k) + \frac{\gamma}{p} e_2(z,k) \right] \tag{35}
$$
$$
+ \sum_{k=0}^{K-1} \left( (12\eta LD + 6\eta D_F + 2D) \|\hat{u}^k - u^k\| + 16\eta^2\delta^2 \|u^k - \hat{u}^k\|^2 \right).
$$

Recall that we take $\hat{c}$ by equation (27), then the setting

$$
\varepsilon_k = \min \left\{ \zeta, \hat{c}^{-1} \min \left\{ \|\hat{u}^k - z^k\|, \|\hat{u}^k - z^k\|^2 \right\} \right\}
$$

satisfies the condition on $\varepsilon_k$ in Lemma 1. Then we apply Lemma 1 with $\mu = 0$ and $\alpha = 1$ and sum over equation (9) with $k = 0, \ldots, K-1$ to obtain

$$
\sum_{k=0}^{K-1} \left( \frac{1}{32} \mathbb{E} \left[ \|z^k - \hat{u}^k\|^2 \right] + \frac{\gamma}{2} \mathbb{E} \left[ \|w^k - \hat{u}^k\|^2 \right] \right) \le \left( 1 + \frac{\gamma}{p} \right) \|z^0 - z^*\|^2. \tag{36}
$$

Note that parameter settings $\gamma = p = 1/(\sqrt{n} + 8)$, $b = \lceil \sqrt{n} \rceil$, and $\eta = \min \left\{ \sqrt{\gamma b}/(4\delta), 1/(32\delta) \right\}$ satisfy

$$
\frac{4\eta^2\delta^2}{b} \le \frac{\gamma}{4} \le 8 \cdot \frac{1}{32}, \quad 2p + \frac{4\eta^2\delta^2}{b} \le 2p + \frac{4\delta^2}{b} \frac{\gamma b}{16\delta^2} \le 5 \cdot \frac{\gamma}{2} \quad \text{and} \quad 1 + \frac{\gamma}{p} = 2. \tag{37}
$$

Substituting equations (36) and (37) into equation (35) and applying Lemma 10, we obtain

$$
\mathbb{E} \left[ \max_{z \in \mathcal{Z}} \frac{1}{K} \sum_{k=0}^{K-1} \langle F(u^k), u^k - z \rangle \right]
$$
$$
\le \frac{1}{2\eta K} (4 + 8 \cdot 2) \max_{z \in \mathcal{Z}} \|z^0 - z\|^2
$$
$$
+ \frac{1}{2\eta K} \sum_{k=0}^{K-1} \left( (12\eta LD + 6\eta D_F + 2D) \|\hat{u}^k - u^k\| + \left( 16\eta^2\delta^2 + \frac{8\eta^2\delta^2}{b} + 2p \right) \|u^k - \hat{u}^k\|^2 \right)
$$
$$
\le \frac{10D^2}{\eta K} + \frac{6\eta LD + 3\eta D_F + D}{\eta} \sqrt{\zeta} + \frac{12\eta^2\delta^2 + 1}{\eta} \zeta
$$
$$
\le \frac{10D^2}{\eta K} + \frac{9\eta LD + 3\eta \max_{i \in [n]} \|F_i(z^0)\| + D}{\eta} \sqrt{\zeta} + \frac{12\eta^2\delta^2 + 1}{\eta} \zeta,
$$

where we use the equation (28) to bound $D_F$.

Recall that we take constant $\zeta$ by equation (26), then we get the bound

$$
\mathbb{E} \left[ \max_{z \in \mathcal{Z}} \frac{1}{K} \sum_{k=0}^{K-1} \langle F(u^k), u^k - z \rangle \right] \le \frac{10D^2}{\eta K} + \frac{\varepsilon}{2}.
$$

$\square$

### C.3 Proof of Corollary 1

*Proof.* Theorem 1 means we can achieve $\mathbb{E}[\mathrm{Gap}(u_{\mathrm{avg}}^K)] \leq \varepsilon$ by taking the communication rounds of

$$K = \left\lceil \frac{20D^2}{\varepsilon\eta} \right\rceil = \mathcal{O}\left(\frac{D^2}{\varepsilon\eta}\right) = \mathcal{O}\left(\frac{\delta D^2}{\varepsilon}\right).$$

Consider that the expected communication complexity in each round is $\mathcal{O}(b(1-p) + np) = \mathcal{O}(\sqrt{n})$ and the server need to communicate with all client in initialization within the communication complexity of $\mathcal{O}(n)$, the overall communication complexity is

$$\mathcal{O}(n) + K \cdot \mathcal{O}(\sqrt{n}) = \mathcal{O}\left(n + \frac{\sqrt{n}\delta D^2}{\varepsilon}\right).$$

Note that the objective of the sub-problem in Line 8 of Algorithm 1 is $(L + 1/\eta)$-smooth-$(1/\eta)$-strongly-convex-$(1/\eta)$-strongly-concave, the local gradient complexity for solving the sub-problem is $\mathcal{O}((1 + \eta L) \log(\max\{\zeta^{-1}, \hat{c}\}))$. Therefore, the overall local gradient complexity is

$$\mathcal{O}(n) + K \cdot \left(\mathcal{O}(\sqrt{n}) + \mathcal{O}\left((1 + \eta L) \log(\zeta^{-1} + \hat{c})\right)\right)$$

$$= \mathcal{O}(n) + \mathcal{O}\left(\frac{\delta D^2}{\varepsilon}\right) \cdot \left(\mathcal{O}(\sqrt{n}) + \mathcal{O}\left(\left(1 + \frac{L}{\delta}\right) \log\left(\frac{LD + D_F}{\varepsilon} + \sqrt{\frac{\Phi^0}{\delta}}\right)\right)\right)$$

$$= \tilde{\mathcal{O}}\left(n + \frac{(\sqrt{n}\delta + L)D^2}{\varepsilon} \log\frac{1}{\varepsilon}\right).$$

$\square$

### C.4 Proof of Theorem 2

*Proof.* We can verify that the parameter setting of Theorem 2 satisfies the condition of Lemma 1. Then we can apply Lemma 1 to obtain

$$\mathbb{E}[\Phi^K] \leq \max\left\{1 - \frac{\eta\mu}{6(1-\gamma)}, 1 - \frac{p\eta\mu}{2\gamma + \eta\mu}\right\}^K \Phi^0. \tag{38}$$

$\square$

### C.5 Proof of Corollary 2

*Proof.* Recall that we set the parameters as

$$\gamma = p = \frac{1}{\min\left\{\sqrt{n}, \frac{\delta}{\mu}\right\} + 8}, \quad b = \left\lceil \min\left\{\sqrt{n}, \frac{\delta}{\mu}\right\} \right\rceil,$$

$$\eta = \min\left\{\frac{\sqrt{\gamma b}}{4\delta}, \frac{1}{32\delta}\right\}, \quad \alpha = \max\left\{1 - \frac{\eta\mu}{6(1-\gamma)}, 1 - \frac{p\eta\mu}{2\gamma + \eta\mu}\right\}.$$

We can lower bound $\alpha$ as

$$\alpha \geq 1 - \frac{p\eta\mu}{2\gamma + \eta\mu} = 1 - \frac{p\eta\mu}{2p + \eta\mu} = 1 - \frac{1}{\frac{2}{\eta\mu} + \frac{1}{p}} \geq \frac{7}{8}. \tag{39}$$

Then the number of communication rounds is

$$K = \mathcal{O}\left(\left(1 + \frac{1}{\eta\mu} + \frac{\gamma + \eta\mu}{p\eta\mu}\right) \log\frac{1}{\varepsilon}\right)$$

$$= \mathcal{O}\left(\left(\frac{1}{p} + \frac{1}{\eta\mu}\right) \log\frac{1}{\varepsilon}\right)$$

$$= \mathcal{O}\left(\left(\frac{1}{p} + \frac{1}{\mu}\left(32\delta + \frac{32\delta}{\sqrt{\alpha\gamma b}}\right)\right) \log\frac{1}{\varepsilon}\right)$$

$$= \mathcal{O}\left(\left(\frac{1}{p} + \frac{\delta}{\mu} + \frac{1}{\sqrt{pb}}\frac{\delta}{\mu}\right) \log\frac{1}{\varepsilon}\right).$$

Note that

$$\frac{1}{p} + \frac{1}{\sqrt{pb}}\frac{\delta}{\mu} \le \min\left\{\sqrt{n}, \frac{\delta}{\mu}\right\} + 8 + \frac{\delta}{\mu}\sqrt{\frac{\min\left\{\sqrt{n}, \frac{\delta}{\mu}\right\} + 8}{\min\left\{\sqrt{n}, \frac{\delta}{\mu}\right\}}}$$

$$= \min\left\{\sqrt{n}, \frac{\delta}{\mu}\right\} + 8 + \frac{\delta}{\mu}\sqrt{1 + 8\max\left\{\frac{1}{\sqrt{n}}, \frac{\mu}{\delta}\right\}}$$

$$= \mathcal{O}\left(\frac{\delta}{\mu}\right),$$

then we have $K = \mathcal{O}(\delta/\mu \log(1/\varepsilon))$.

Consider that the expected communication complexity in each round is

$$\mathcal{O}(b(1 - p) + np) = \mathcal{O}\left(\sqrt{n} + \frac{n\mu}{\delta}\right),$$

and the server need to communicate with all client in initialization within the communication complexity of $\mathcal{O}(n)$, the overall communication complexity is

$$\mathcal{O}(n) + K \cdot \mathcal{O}\left(\sqrt{n} + \frac{n\mu}{\delta}\right) = \mathcal{O}\left(\left(n + \frac{\sqrt{n}\delta}{\mu}\right)\log\frac{1}{\varepsilon}\right).$$

Note that the objective of the sub-problem in Line 8 of Algorithm 1 is $(L + 1/\eta)$-smooth-$(1/\eta)$-strongly-convex-$(1/\eta)$-strongly-concave, the local gradient complexity for solving the sub-problem is $\mathcal{O}((1 + \eta L)\log(c))$. Therefore, the overall local gradient complexity is

$$\mathcal{O}(n) + K \cdot \left(\mathcal{O}\left(\sqrt{n} + \frac{n\mu}{\delta}\right) + \mathcal{O}\left((1 + \eta L)\log(c)\right)\right)$$

$$= \mathcal{O}(n) + \mathcal{O}\left(\frac{\delta}{\mu}\log\frac{1}{\varepsilon}\right) \cdot \left(\mathcal{O}\left(\sqrt{n} + \frac{n\mu}{\delta}\right) + \mathcal{O}\left(\left(1 + \frac{L}{\delta}\right)\log\frac{\delta}{\mu}\right)\right)$$

$$= \mathcal{O}\left(\left(n + \frac{\sqrt{n}\delta}{\mu} + \frac{L}{\mu}\log\frac{\delta}{\mu}\right)\log\frac{1}{\varepsilon}\right)$$

$$= \tilde{\mathcal{O}}\left(\left(n + \frac{\sqrt{n}\delta + L}{\mu}\right)\log\frac{1}{\varepsilon}\right).$$

$\square$

## D  The Algorithm Class

We formally define the distributed first-order oracle (DFO) algorithm as follows.

**Definition 1** (DFO Algorithm). *Each node $i$ has its own local memories $\mathcal{M}_i^x$ and $\mathcal{M}_i^y$ for the $x$- and $y$-variables with initialization $\mathcal{M}_i^x = \mathcal{M}_i^y = \{0\}$ for all $i \in [n]$. Specifically, the server has memories $\mathcal{M}_1^x$ and $\mathcal{M}_1^y$. These memories $\{\mathcal{M}_i^x\}_{i=1}^n$ and $\{\mathcal{M}_i^y\}_{i=1}^n$ can be updated as follows:*

- ***Communication from clients to server****: During one communication round, we sample uniformly and independently batch $\mathcal{S}$ of any size $b$ and ask client with number from $\mathcal{S}$ to share some vector of their local memories with the server, i.e. can add points $x_1', y_1'$ to the local memories of the server according to the next rule:*

$$x_1' \in \text{span}\left\{x_1, \bigcup_{i \in \mathcal{S}} x_i\right\} \qquad and \qquad y_1' \in \text{span}\left\{y_1, \bigcup_{i \in \mathcal{S}} y_i\right\}$$

*where $x_i \in \mathcal{M}_i^x$ and $y_i \in \mathcal{M}_i^y$. If the batch size is equal to $b$ we say that it costs $b$ communication complexity from clients to the server. Batch of the size $n$ is equal to the situation, when all clients send their memories to the server.*

- ***Communication from server to clients****: During one communication round, we sample uniformly and independently batch $\mathcal{S}$ of any size $b$ and ask the server to share some vector of its local*

*memories with the clients with numbers from $\mathcal{S}$, i.e. can add points $x'_i, y'_i$ to the corresponding local memories of client $i$ as*

$$x'_i \in \text{span}\{x_1, x_i\} \qquad and \qquad y'_i \in \text{span}\{y_1, y_i\},$$

*where $x_i \in \mathcal{M}_i^x$ and $y_i \in \mathcal{M}_i^y$, and we say that it costs $b$ communication complexity.*

- **Local computations**: *During local computations each client $i$ can make any computations using $f_i$, i.e. can add points $x'_i, y'_i$ to the corresponding local memory of client $i$ as*

$$x'_i \in \text{span}\{x', \nabla_x f_i (x'', y'')\} \qquad and \qquad y'_i \in \text{span}\{y', \nabla_y f_i (x'', y'')\},$$

*for given $x', x'' \in \mathcal{M}_i^x$ and $y', y'' \in \mathcal{M}_i^y$. And we use local gradient calls to count the times when $\nabla_x$ and $\nabla_y$ are applied to any one of $\{f_i\}$.*

*The final global output is calculated as $\hat{x} \in \mathcal{M}_1^x, \hat{y} \in \mathcal{M}_1^y$.*

Our Definition 1 follows the algorithm class of Beznosikov et al. [8, Definition C.7], but additionally take the communication from the server to the clients into considerations.

# E  Proofs for Lower Bounds in Convex-Concave Case

In this section, we provide the proofs of the lower bounds for solving the problem

$$\min_{x \in \mathcal{X}} \max_{y \in \mathcal{Y}} f(x, y) = \frac{1}{n} \sum_{i=1}^{n} f_i(x, y) \tag{40}$$

by DFO algorithms, where the diameters of closed convex sets $\mathcal{X}$ and $\mathcal{Y}$ are $R_x$ and $R_y$ respectively. We define the subspaces $\{\mathcal{F}_k\}_{k=0}^{d}$ as

$$\mathcal{F}_k = \begin{cases} \text{span}\{e_1, \ldots, e_k\}, & \text{for } 1 \leq k \leq d, \\ \{0_d\}, & \text{for } k = 0, \end{cases}$$

which is used in the following proofs of lower bounds.

## E.1  Proof of Theorem 3

We first define the function set with one server ($i = 1$) and $n - 1$ clients ($i = 2, \ldots, n-1$) as follows

$$f_i(x, y) = \begin{cases} \dfrac{\delta}{4} x^\top A_1 y - \dfrac{\delta R_y}{2\sqrt{d}} e_1^\top x, & i - 1 \equiv 1 \pmod 3, \\[2mm] \dfrac{\delta}{4} x^\top A_2 y, & i - 1 \equiv 2 \pmod 3, \\[2mm] 0, & \text{otherwise.} \end{cases} \tag{41}$$

Then corresponding global objective is

$$f(x, y) = \frac{\delta}{6} x^\top A y - \frac{\delta R_y}{6\sqrt{d}} e_1^\top x, \tag{42}$$

where

$$A_1 = \begin{pmatrix} 1 & 0 & & & \\ & 1 & -2 & & \\ & & \ddots & \ddots & \\ & & & 1 & 0 \\ & & & & 1 \end{pmatrix}, \ A_2 = \begin{pmatrix} 1 & -2 & & & \\ & 1 & 0 & & \\ & & \ddots & \ddots & \\ & & & 1 & -2 \\ & & & & 1 \end{pmatrix}, \ A = \begin{pmatrix} 1 & -1 & & & \\ & 1 & -1 & & \\ & & \ddots & \ddots & \\ & & & 1 & -1 \\ & & & & 1 \end{pmatrix}. \tag{43}$$

**Proposition 2.** *For any $d \geq 3$, the functions $f_i(x, y)$ and $f(x, y)$ defined by equations (41) and (42) satisfy*

1. *The function $f_i$ is L-smooth with $L \geq \delta$ and convex-concave for all $i \in [n]$, and the function set $\{f_i\}_{i=1}^n$ holds $\delta$-second-order similarity. Thus, the function $f$ is also convex-concave.*

2. *For $1 \leq k \leq d - 1$, we have*

$$\min_{(x,y) \in \mathcal{Z} \cap \mathcal{F}_k^2} \mathrm{Gap}(x,y) = \min_{x \in \mathcal{X} \cap \mathcal{F}_k} \max_{y \in \mathcal{Y}} f(x,y) - \max_{y \in \mathcal{Y} \cap \mathcal{F}_k} \min_{x \in \mathcal{X}} f(x,y) \geq \frac{\delta R_x R_y}{6\sqrt{d(k+1)}}. \quad (44)$$

*Proof.* The smoothness and the convexity (concavity) are easy to verify. The similarity holds because

$$\nabla_{xx}^2 f_1(x,y) - \nabla_{xx}^2 f(x,y) = \nabla_{xx}^2 f_2(x,y) - \nabla_{xx}^2 f(x,y) = \nabla_{xx}^2 f_3(x,y) - \nabla_{xx}^2 f(x,y) = 0;$$

$$\nabla_{yy}^2 f_1(x,y) - \nabla_{yy}^2 f(x,y) = \nabla_{yy}^2 f_2(x,y) - \nabla_{yy}^2 f(x,y) = \nabla_{yy}^2 f_3(x,y) - \nabla_{yy}^2 f(x,y) = 0;$$

$$\|\nabla_{xy}^2 f_1(x,y) - \nabla_{xy}^2 f(x,y)\| \leq \|\nabla_{xy}^2 f_1(x,y)\| + \|\nabla_{xy}^2 f(x,y)\| \leq \frac{\delta}{3} \leq \delta;$$

$$\|\nabla_{xy}^2 f_2(x,y) - \nabla_{xy}^2 f(x,y)\| \leq \|\nabla_{xy}^2 f_2(x,y)\| + \|\nabla_{xy}^2 f(x,y)\| \leq \delta \left(\frac{5}{8} + \frac{1}{3}\right) \leq \delta;$$

$$\|\nabla_{xy}^2 f_3(x,y) - \nabla_{xy}^2 f(x,y)\| \leq \|\nabla_{xy}^2 f_3(x,y)\| + \|\nabla_{xy}^2 f(x,y)\| \leq \delta \left(\frac{5}{8} + \frac{1}{3}\right) \leq \delta.$$

The function $f(x,y)$ defined by our equation (42) is identical to the function $f_{\mathrm{CC}}(x,y)$ defined by Han et al. [14, Proposition 3.31][3] by replacing their notation $L$ with our $\delta$ and taking $n = 3$. Then Proposition 3.31 of Han et al. [14] directly prove the result of equation (44). $\qquad \square$

The structure of $A_1$ and $A_2$ results the following lemma.

**Lemma 11.** *For the function set (41), all $(x,y) \in \mathcal{F}_k \times \mathcal{F}_k$ and $k = 0, \ldots, d-1$, we have*

$$\nabla f_i(x,y) \in \begin{cases} \mathcal{F}_{k+1} \times \mathcal{F}_{k+1}, & (i,k) \in \mathcal{I}_1 \cup \mathcal{I}_2, \\ \mathcal{F}_k \times \mathcal{F}_k, & \textit{otherwise}, \end{cases}$$

*where the index sets are defined as $\mathcal{I}_1 := \{(i,k) : i - 1 \equiv 1 \pmod 3, k \equiv 0 \pmod 2\}$ and $\mathcal{I}_2 := \{(i,k) : i - 1 \equiv 2 \pmod 3, k \equiv 1 \pmod 2\}$.*

Now we provide the proof of Theorem 3.

*Proof.* Consider the minimax problem (40) with functions (41) and (42), $R_x = R_y = D$, $n \geq 3$ and $d = \lfloor \delta D^2/(3\sqrt{2}\varepsilon) \rfloor - 1$. Then the assumption $\varepsilon \leq \delta D^2/(12\sqrt{2})$ implies $d \geq 3$. Lemma 11 means that we need at least one communication round to increase the number of non-zero coordinate, i.e. $(x,y) \in \mathcal{Z} \cap \mathcal{F}_K^2$. Running any DFO algorithm with communication rounds of $K = \lfloor (d-1)/2 \rfloor \geq 1$, we have $d/2 \leq (K+1) \leq (d+1)/2$ and Proposition 2 implies

$$\mathbb{E}[\mathrm{Gap}(x,y)] \geq \min_{(x,y) \in \mathcal{Z} \cap \mathcal{F}_K^2} \mathrm{Gap}(x,y) \geq \frac{\delta D^2}{6\sqrt{d(K+1)}} \geq \frac{\delta D^2}{6\sqrt{2}(K+1)} \geq \frac{\delta D^2}{3\sqrt{2}(d+1)} \geq \varepsilon.$$

Hence, we achieve the lower bound on the communication rounds of

$$K = \left\lfloor \frac{\delta D^2}{6\sqrt{2}\varepsilon} \right\rfloor - 1 = \Omega\left(\frac{\delta D^2}{\varepsilon}\right).$$

$\qquad \square$

### E.2 Proof of Theorem 4

We first define the function set with one server ($i = 1$) and $n - 1$ clients ($i = 2, \ldots, n - 1$) as follows

$$f_i(x,y) = \begin{cases} -\dfrac{\sqrt{n}\delta R_y}{8\sqrt{d}} e_1^\top x, & i = 1, \\[4mm] \dfrac{\sqrt{n}\delta}{8} x^\top \left[ \displaystyle\sum_{j \equiv (i-1) \bmod (n-1)} e_j a_j^\top \right] y, & i \geq 2. \end{cases} \quad (45)$$

---

[3]We follow the notation of Han et al. [14]'s arXiv version: `https://arxiv.org/pdf/2103.08280v1`

Then corresponding global objective is

$$f(x,y) = \frac{\delta}{8\sqrt{n}} x^\top A y - \frac{\delta R_y}{8\sqrt{nd}} e_1^\top x, \tag{46}$$

where $e_j$ is the $j$-th basis column vector, $a_j^\top$ is the $j$-th row of $A$, $A_i = \sum_{j \equiv (i-1) \bmod (n-1)} e_j a_j^\top$, $A_1 = 0$ and $A$ is defined in equation (43).

We provide the following proposition and lemmas for the proof of Theorem 4.

**Proposition 3.** *For any $d \geq 3$, $f_i(x,y)$ and $f(x,y)$ defined as equations (45) and (46) satisfy*

1. *$f_i$ is $L$-smooth with $L \geq \sqrt{n}\delta/4$ and convex-concave, function set $\{f_i\}_{i=1}^n$ has $\delta$ second-order similarity. Thus, $f$ is convex-concave.*

2. *For $1 \leq k \leq d-1$, we have*

$$\min_{(x,y)\in\mathcal{Z}\cap\mathcal{F}_k^2} \mathrm{Gap}(x,y) = \min_{x\in\mathcal{X}\cap\mathcal{F}_k} \max_{y\in\mathcal{Y}} f(x,y) - \max_{y\in\mathcal{Y}\cap\mathcal{F}_k} \min_{x\in\mathcal{X}} f(x,y) \geq \frac{\delta R_x R_y}{8\sqrt{nd(k+1)}}. \tag{47}$$

*Proof.* The smoothness and convexity (concavity) are easy to check. And the similarity can be verified following the methods of Beznosikov et al. [8, Lemma C.8]. The function $f(x,y)$ defined by our equation (46) is identical to the function $f_{CC}(x,y)$ defined by Han et al. [14, Proposition 3.31] by replacing their notation $L$ with our $\sqrt{n}\delta/4$. Then Proposition 3.31 of Han et al. [14] directly prove the result of equation (47). □

**Lemma 12.** *Consider the minimax problem (40) with functions (45) and (46), $R_x = R_y = D$, $d = \lfloor \delta D^2/(4\sqrt{2n}\varepsilon) \rfloor - 1$ and $\varepsilon \leq \delta D^2/(16\sqrt{2n})$. We let $M = \lfloor (d-1)/2 \rfloor \geq \Omega(\delta D^2/(\sqrt{n}\varepsilon))$, then any point $(x,y) \in \mathcal{Z} \cap \mathcal{F}_M^2$ satisfies $\mathrm{Gap}(x,y) \geq \varepsilon$.*

*Proof.* The assumptions on $\varepsilon$ and $M$ imply $d \geq 3$. and $d/2 \leq (M+1) \leq (d+1)/2$. Then Proposition 3 means

$$\mathrm{Gap}(x,y) \geq \min_{(x,y)\in\mathcal{Z}\cap\mathcal{F}_M^2} \mathrm{Gap}(x,y) \geq \frac{\delta D^2}{8\sqrt{nd(M+1)}} \geq \frac{\delta D^2}{8\sqrt{2n}(M+1)} \geq \frac{\delta D^2}{4\sqrt{2n}(d+1)} \geq \varepsilon.$$

□

**Lemma 13.** *Consider the minimax problem (40) with functions (45) and (46) and run any DFO algorithm with $V$ communication complexity and $C$ local gradient calls. In expectation, only the first $M \leq \min\{2V/n, 2C/n\}$ coordinates of the final output can be non-zero while the rest of the $d - M$ coordinates are strictly equal to zero.*

*Proof.* At initialization, $\mathcal{M}_i^x = \mathcal{M}_i^y = \mathcal{F}_0$. Let's analyze how $\mathcal{M}_i^x$ and $\mathcal{M}_i^y$ change through local computations. For the $i$-th client, we add the following points to $\mathcal{M}_i^x$ and $\mathcal{M}_i^y$ as

$$x \in \mathrm{span}\left\{x', A_i y'\right\}, \quad \text{and} \quad y \in \mathrm{span}\left\{e_1 \cdot \mathbb{I}\{i=1\}, y', A_i^\top x'\right\},$$

where $x' \in \mathcal{M}_i^x$ and $y' \in \mathcal{M}_i^y$.

It is easy to see that the server can make the first coordinate of $y$ non-zero using $e_1$, and broadcast this progress to other clients. Only updates of the type $A_i y'$ or $A_i^\top x'$ will help in this regard. Since $A_i$ only contains rows from the matrix $A$ such as the $(i-1)$-th row, $(n+i-1)$-th row, etc., to make the first coordinate of $x$ in the global output non-zero, we need and can only use the $A_2$ matrix. It can be noted that by using $A_2$, we can also make the second coordinate of $y$ non-zero after making the first coordinate of $x$ non-zero. Furthermore, to make more progress, we need to use $A_3$ and so on. We conclude that we must constantly transfer progress from the node currently needed (to make the next coordinate of $x$ non-zero; then of $y$) to the server, and then to other nodes.

By definition, one communication round involves communication with all clients or only with batches of some uniform and independent clients. When we sample without replacement, the success probability of a communication round on clients with batch size $b$ (i.e., making one coordinate

non-zero) is $\frac{b}{n-1}$ (or 1 when $b = n$), which is also the expected number of non-zero coordinates that can be obtained with $b$ communication complexity and at least $b$ local gradient calls (as each use of the matrix $A$ necessarily comes from a gradient call). When we sample with replacement by a batch size $b$, it is equivalent to that we sample without replacement by batch size 1 for $b$ times. Assuming that we have communication rounds with batch sizes (sampling without replacement) of $1, 2, \ldots, n$ for $s_1, s_2, \ldots, s_n$ times, then the communication complexity we spent is $V = \sum_{j=1}^{n} j s_j$ and the minimum gradient calls we spent is $C = \sum_{j=1}^{n} j s_j$. This implies the expected number of non-zero coordinates is $M = \sum_{j=1}^{n-1} \frac{j}{n-1} s_j + s_n$. Therefore, the expected total number of non-zero coordinates in the global output is at most $M$ for $y$ and $M-1$ for $x$ (or we can say $M$). By comparing expressions of $V$, $C$ and $M$, we can have $M \leq 2V/n$ and $M \leq 2C/n$, completing the proof. □

Then we can prove Theorem 4 by combining Lemma 12 and 13.

### E.3 Proof of Lemma 2

We choose the function set as

$$f(x, y) = f_i(x, y) = \frac{L}{2} x^\top A y - \frac{L R_y}{2\sqrt{d}} e_1^\top x, \quad \forall i \in [n], \tag{48}$$

where $A$ is defined as equation (43). The structure of $A$ results the following lemma.

**Lemma 14.** *For the function set (48), all $(x, y) \in \mathcal{F}_k \times \mathcal{F}_k$ and $k = 0, \ldots, d-1$, we have*

$$\nabla f_i(x, y) \in \mathcal{F}_{k+1} \times \mathcal{F}_{k+1}.$$

One can follow the similar method as the proof of Theorem 4 to prove the following proposition and lemma for the proof of Lemma 2.

**Proposition 4.** *For any $d \geq 3$, $f_i(x, y)$ and $f(x, y)$ defined as equation (48) satisfy*

1. *$f_i$ is $L$-smooth and convex-concave, function set $\{f_i\}_{i=1}^{n}$ has $\delta$ second-order similarity for any $\delta > 0$. Thus, $f$ is convex-concave.*

2. *For $1 \leq k \leq d-1$, we have*

$$\min_{(x,y) \in \mathcal{Z} \cap \mathcal{F}_k^2} \mathrm{Gap}(x, y) = \min_{x \in \mathcal{X} \cap \mathcal{F}_k} \max_{y \in \mathcal{Y}} f(x, y) - \max_{y \in \mathcal{Y} \cap \mathcal{F}_k} \min_{x \in \mathcal{X}} f(x, y) \geq \frac{L R_x R_y}{2\sqrt{d(k+1)}}.$$

**Lemma 15.** *Consider the minimax problem (40) with the function set (48), $R_x = R_y = D$, $d = \lfloor \delta D^2/(\sqrt{2}\varepsilon) \rfloor - 1$ and $\varepsilon \leq \delta D^2/(4\sqrt{2})$. We let $M = \lfloor (d-1)/2 \rfloor \geq \Omega(L D^2/\varepsilon)$, then any point $(x, y) \in \mathcal{Z} \cap \mathcal{F}_M^2$ satisfies $\mathrm{Gap}(x, y) \geq \varepsilon$.*

Then we can prove Lemma 2 by combining Lemma 14 and 15.

### E.4 Proof of Theorem 5

*Proof.* In the case of $\sqrt{n}\delta \geq \Omega(L)$, the problem with with functions (45) and (46) in Theorem 4 implies the lower bound on the local gradient complexity of

$$\Omega\left(n + \frac{\sqrt{n}\delta D^2}{\varepsilon}\right) = \Omega\left(n + \frac{(\sqrt{n}\delta + L)D^2}{\varepsilon}\right).$$

In the case of $L \geq \Omega(\sqrt{n}\delta)$, the problem with function (48) in Lemma 2 implies the lower bound on the local gradient complexity of $\Omega(LD^2/\varepsilon) = \Omega(n + (\sqrt{n}\delta + L)D^2/\varepsilon)$. Combining these two cases, we achieve the lower bound on the local gradient complexity of $\Omega(n + (\sqrt{n}\delta + L)D^2/\varepsilon)$. □

## F  Proofs for Lower Bounds in Strongly-Convex-Strongly-Concave Case

We follow similar steps as in Appendix E to prove Theorem 7. Firstly we divide it into several detailed theorems and lemmas as below and prove them one by one.

**Theorem 8.** *For any $\mu, \delta, L > 0$ with $L \geq \max\{\mu, \delta\}$ and $n \geq 2$, there exist $L$-smooth and convex-concave functions $f_1, \ldots, f_n : \mathbb{R}^{d_x} \times \mathbb{R}^{d_y}$ with $\delta$-second-order similarity such that the function $f(x, y) = \frac{1}{n} \sum_{i=1}^{n} f_i(x, y)$ is $\mu$-strongly-convex-$\mu$-strongly-concave. In order to find a solution of problem (1) such that $\mathbb{E}[\|z - z^*\|^2] \leq \varepsilon$, any DFO algorithm needs at least $\Omega((n + \sqrt{n}\delta/\mu) \log(1/\varepsilon))$ communication complexity and $\Omega((n + \sqrt{n}\delta/\mu) \log(1/\varepsilon))$ local gradient calls.*

**Lemma 16.** *For any $\mu, \delta, L > 0$ with $L \geq \max\{\mu, \delta\}$ and $n \geq 2$, there exist $L$-smooth and convex-concave functions $f_1, \ldots, f_n : \mathbb{R}^{d_x} \times \mathbb{R}^{d_y}$ with $\delta$-second-order similarity such that the function $f(x, y) = \frac{1}{n} \sum_{i=1}^{n} f_i(x, y)$ is $\mu$-strongly-convex-$\mu$-strongly-concave. In order to find a solution of problem (1) such that $\mathbb{E}[\|z - z^*\|^2] \leq \varepsilon$, any DFO algorithm needs at least $\Omega(L/\mu \log(1/\varepsilon))$ local gradient calls.*

**Theorem 9.** *For any $\mu, \delta, L > 0$ with $L \geq \max\{\mu, \delta\}$ and $n \geq 2$, there exist $L$-smooth and convex-concave functions $f_1, \ldots, f_n : \mathbb{R}^{d_x} \times \mathbb{R}^{d_y}$ with $\delta$-second-order similarity such that the function $f(x, y) = \frac{1}{n} \sum_{i=1}^{n} f_i(x, y)$ is $\mu$-strongly-convex-$\mu$-strongly-concave. In order to find a solution of problem (1) such that $\mathbb{E}[\|z - z^*\|^2] \leq \varepsilon$, any DFO algorithm needs at least $\Omega((n + (\sqrt{n}\delta + L)/\mu) \log(1/\varepsilon))$ local gradient calls.*

### F.1 Proof of Theorem 8

We introduce the function set as in [8], which is similar to equation (45) that

$$
f_i(x, y) = \begin{cases} \dfrac{\mu}{2}\|x\|^2 - \dfrac{\mu}{2}\|y\|^2 + \dfrac{\delta^2}{16\mu}e_1^\top y, & i = 1, \\[2ex] \dfrac{\delta\sqrt{n}}{4}x^\top \left[ \displaystyle\sum_{j \equiv (i-1) \bmod (n-1)} e_j a_j^\top \right] y + \dfrac{\mu}{2}\|x\|^2 - \dfrac{\mu}{2}\|y\|^2, & i > 2. \end{cases} \tag{49}
$$

Then corresponding global objective is

$$
f(x, y) = \frac{\delta}{4\sqrt{n}}x^\top A y + \frac{\mu}{2}\|x\|^2 - \frac{\mu}{2}\|y\|^2 + \frac{\delta^2}{16n\mu}e_1^\top y, \tag{50}
$$

where all the notation keeps the same as the former section. We point out that the global objective function (50) with local functions (49) satisfies Assumptions 1, 3, 4, 5, and 6, with constant $\mu$, $\delta$, and $L \geq \delta$. See Lemma C.8 of Beznosikov et al. [8] for details.

We provide the following lemmas for the proof of Theorem 8.

**Lemma 17.** *Consider the minimax problem (40) with functions (49) and (50) and run any DFO algorithm with $V$ communication complexity and $C$ local gradient calls. In expectation, only the first $M \leq \min\{2V/n, 2C/n\}$ coordinates of the final output can be non-zero while the rest of the $d - M$ coordinates are strictly equal to zero.*

*Proof.* Follow the proof of Lemma 13. $\qquad\square$

**Lemma 18** (Beznosikov et al. [8, Theorem C.10]). *Let $\mu, \delta > 0$, $n \in \mathbb{N}$, $M \in \mathbb{N}$. There exists a centralized distributed saddle-point problem with functions (49) and (50), in which $z^* \neq 0$ and $d \geq \max\{2\log_q(\alpha/(4\sqrt{2})), 2M\}$ where $\alpha = 16n\mu^2/\delta^2$ and $q = (2 + \alpha - \sqrt{\alpha^2 + 4\alpha})/2 \in (0, 1)$. Then for any output $(\hat{x}, \hat{y})$ of any DFO algorithm leaving $d - M$ coordinates zero, one can obtain the following estimate:*

$$
\|\hat{x} - x^*\|^2 + \|\hat{y} - y^*\|^2 = \Omega\left( \exp\left( -\frac{16}{1 + \sqrt{\frac{\delta^2}{16\mu^2 n} + 1}} \cdot M \right) \|y_0 - y^*\|^2 \right).
$$

Then we can prove Theorem 8 by combining Lemma 17 and 18 and noting that to reach a solution of $\varepsilon$-accuracy requires

$$
\min\left\{ \frac{2V}{n}, \frac{2C}{n} \right\} \geq M = \Omega\left( \left( 1 + \frac{\delta}{\sqrt{n}\mu} \right) \log \frac{1}{\varepsilon} \right).
$$

### F.2 Proof of Lemma 16

We introduce the function set as

$$f(x, y) = f_i(x, y) = \frac{L}{2} x^\top \tilde{A} y + \frac{\mu}{2} \|x\|^2 - \frac{\mu}{2} \|y\|^2 + \frac{L^2}{4\mu} e_1^\top y, \quad \forall i \in [n], \qquad (51)$$

where

$$\tilde{A} = \begin{pmatrix} & & & 1 \\ & & 1 & -1 \\ & \cdot^{\cdot^{\cdot}} & \cdot^{\cdot^{\cdot}} & \\ 1 & -1 & & \end{pmatrix}.$$

Note that since all the nodes have the same function, this function set (51) satisfies Assumptions 1, 3, 4, 5, and 6 for $L, \mu$ and any $\delta > 0$. The structure of $\tilde{A}$ results the following lemma.

**Lemma 19.** *For the function set (51), all $(x, y) \in \mathcal{F}_k \times \mathcal{F}_k$ and $k = 0, \dots, d-1$, we have*

$$\nabla f_i(x, y) \in \mathcal{F}_{k+1} \times \mathcal{F}_{k+1}.$$

**Lemma 20.** *Let $\mu, \delta, L > 0$, $n \in \mathbb{N}$, $C \in \mathbb{N}$. There exists a centralized distributed saddle-point problem with functions (51), in which $z^* \neq 0$ and $d \geq \max\{2 \log_q(\alpha/\sqrt{2}), 4C\}$ where $\alpha = \mu^2/L^2$ and $q = 1 + 2\alpha - 2\sqrt{\alpha^2 + \alpha} \in (0, 1)$. Then for any DFO algorithm, the output $(x^C, y^C)$ after $C$ local gradient calls will satisfies*

$$\|y^C - y^*\|^2 \geq q^C \cdot \frac{\|y^0 - y^*\|^2}{16}. \qquad (52)$$

*Proof.* Follow the proof of Theorem 3.5 by Zhang et al. [55], in which we take $L_x = L_y = L_{xy} = L$ and $\mu_x = \mu_y = \mu$. $\qquad \square$

Then we can prove Lemma 16 by combining Lemma 19 and 20 and noting that to reach a solution with $\varepsilon$-accuracy needs at least $\Omega(L/\mu \log(1/\varepsilon))$ local gradient calls from equation (52).

### F.3 Proof of Theorem 9

*Proof.* In the case of $n\mu + \sqrt{n}\delta \geq \Omega(L)$, the problem with with functions (49) and (50) in Theorem 8 implies the lower bound on the local gradient complexity of

$$\Omega\left(\left(n + \frac{\sqrt{n}\delta}{\mu}\right) \log \frac{1}{\varepsilon}\right) = \Omega\left(\left(n + \frac{\sqrt{n}\delta + L}{\mu}\right) \log \frac{1}{\varepsilon}\right).$$

In the case of $L \geq \Omega(n\mu + \sqrt{n}\delta)$, the problem with function (51) in Lemma 16 implies the lower bound on the local gradient complexity of $\Omega(L/\mu \log(1/\varepsilon)) = \Omega((n + (\sqrt{n}\delta + L)/\mu) \log(1/\varepsilon))$. Combining these two cases, we can achieve the lower bound on the local gradient complexity of $\Omega((n + (\sqrt{n}\delta + L)/\mu) \log(1/\varepsilon))$. $\qquad \square$

## G  Making the Gradient Small

In constrained case, we can also use gradient mapping to measure the sub-optimality of a solution $z$, that is

$$\mathcal{F}_\tau(z) = \frac{z - \mathcal{P}_{\mathcal{Z}}(z - \tau F(z))}{\tau}.$$

For the $L$-smooth convex-concave function $f$, we consider the constrained problem

$$\min_{x \in \mathcal{X}} \max_{y \in \mathcal{Y}} \hat{f}(x, y) := f(x, y) + \frac{\lambda}{2} \|x - x^0\|^2 - \frac{\lambda}{2} \|y - y^0\|^2,$$

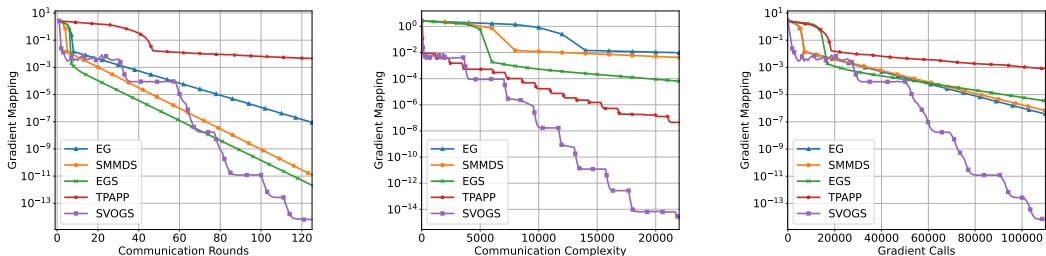

Figure 5: Results for convex-concave minimax problem (12) on covtype.

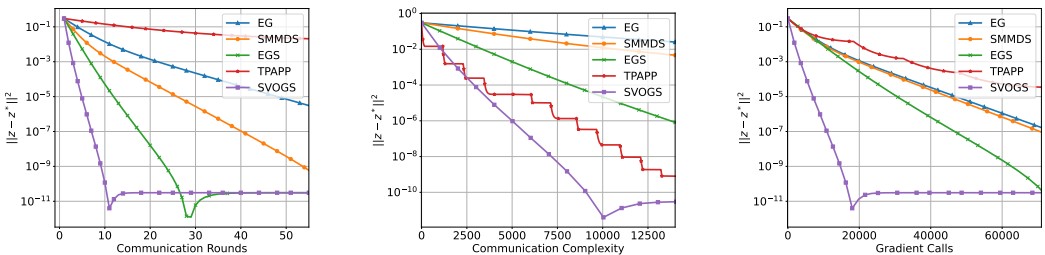

Figure 6: Results for strongly-convex-strongly-concave minimax problem (13) on covtype.

where $\hat{f}$ is $\lambda$-strongly-convex-$\lambda$-strongly-concave and $\mathcal{Z}$ is bounded by diameter $D$. From the the result of Theorem 2, we have $\mathbb{E}\big[\,\big\|z^K - \hat{z}^*\big\|^2\,\big] \leq (1 - \chi\lambda/\delta)^K \big\|z^0 - \hat{z}^*\big\|^2$ for some constant $\chi \in (0,1)$, then we have

$$\mathbb{E}\left[\|\mathscr{F}_\tau(z^K)\|\right] = \mathbb{E}\left\|\frac{z^K - \mathcal{P}_{\mathcal{Z}}\left(z^K - \tau(\hat{F}(z^K) - \lambda(z^K - z^0))\right)}{\tau}\right\|$$

$$= \mathbb{E}\left\|\frac{\mathcal{P}_{\mathcal{Z}}(z^K) - \mathcal{P}_{\mathcal{Z}}\left(z^K - \tau(\hat{F}(z^K) - \lambda(z^K - z^0))\right)}{\tau}\right\|$$

$$\leq \mathbb{E}\left[\left\|\hat{F}(z^K) - \lambda(z^K - z^0)\right\|\right]$$

$$\leq \mathbb{E}\left[\left\|\hat{F}(z^K)\right\|\right] + \lambda\mathbb{E}\left[\left\|z^K - z^0\right\|\right]$$

$$= \mathbb{E}\left[\left\|\hat{F}(z^K) - \hat{F}(\hat{z}^*)\right\|\right] + \lambda\mathbb{E}\left[\left\|z^K - z^0\right\|\right]$$

$$\leq L\mathbb{E}\left[\left\|z^K - \hat{z}^*\right\|\right] + \lambda\mathbb{E}\left[\left\|z^K - z^0\right\|\right],$$

where we use the property of the projection that $\|\mathcal{P}_{\mathcal{Z}}(z_1) - \mathcal{P}_{\mathcal{Z}}(z_2)\| \leq \|z_1 - z_2\|$ and the gradient mapping holds that $\mathscr{F}_\tau(\hat{z}^*) = 0$. Then we have

$$\mathbb{E}\left[\left\|\mathscr{F}(z^K)\right\|^2\right] \leq 2L^2\mathbb{E}\left[\left\|z^K - \hat{z}^*\right\|^2\right] + 2\lambda^2\mathbb{E}\left[\left\|z^K - z^0\right\|^2\right]$$

$$\leq 2L^2(1 - \chi\lambda/\delta)^K \left\|z^0 - \hat{z}^*\right\|^2 + 2\lambda^2\mathbb{E}\left[\left\|z^K - z^0\right\|^2\right]$$

$$\leq (2L^2(1 - \chi\lambda/\delta)^K + 2\lambda^2)D^2.$$

Let $\lambda = \sqrt{\varepsilon/(4D^2)}$ and $K = \mathcal{O}(\delta/\lambda \log(L/\varepsilon))$, then we have $\mathbb{E}\big[\,\big\|\mathscr{F}(z^K)\big\|^2\,\big] \leq \varepsilon$.

Hence, the complexity of communication rounds is $K = \mathcal{O}\left(\delta D/\sqrt{\varepsilon}\log(L/\varepsilon)\right)$. We verify that the corresponding communication complexity is $\tilde{\mathcal{O}}((n + \sqrt{n}\delta D/\sqrt{\varepsilon})\log(1/\varepsilon))$ and the local gradient complexity is $\tilde{\mathcal{O}}((n + (\sqrt{n}\delta + L)D/\sqrt{\varepsilon})\log(1/\varepsilon))$ by following the discussion in Appendix C.5.

# H More Experimental Results

We present the experimental results on dataset "covtype" in Figure 5 and 6.

