# OpenReview forum: "Near-Optimal Distributed Minimax Optimization under the Second-Order Similarity"
_NeurIPS.cc/2024/Conference — NeurIPS 2024 poster_

### Official Review · Reviewer_DkHm · 2024-07-08

**Soundness:** 2
**Presentation:** 3
**Contribution:** 3
**Rating:** 5
**Confidence:** 4

**Summary:**

The paper proposes SVOGS, and improved algorithm for distributed minimax optimization by client mini-batch sampling and gradient variane reduction. Theoretical rates on communication complexity and gradient computations are provided, along with their lower bounds. The analysis shows that SVOGS achieves the corresponding lower bounds. A numerical example is provided to justify the efficacy.

**Strengths:**

1. The presentation is clear and the notations are clean.

2. The paper is theoretically solid, with comprehensive thereotical results. The lower bounds, although based on some tricks and results from prior works, provide novel contribution to the field of minimax optimization.

**Weaknesses:**

The main drawback is the experiment section. The proposed method is only tested on one small dataset, and the paper lacks the basic information on how SVOGS is implemented and tuned (e.g., parameter $b, p, \alpha, \gamma$ in Algorithm 1). It seems to me that the proposed method requires much heavier fint-tuning than prior methods thus less practically convenient. The implementation details should be provided to justify the true practical value of this method.

Also, it should be tested on more datasets.

**Questions:**

See above

---

> ### Author Rebuttal · Authors · 2024-08-03
>
> **The implementation details**
>
> We tune the step-size $\eta$ of SVOGS from $\\{0.01,0.1,1\\}$. The probability $p$ is tuned from $\\{p_0,5p_0,10p_0\\}$, where $p_0=1/\min\\{\sqrt{n}+\delta/\mu\\}$. The batch size $b$ is determined from $\\{\lfloor b_0/10\rfloor,\lfloor b_0/5\rfloor,\lfloor b_0\rfloor\\}$, with $b_0=1/p_0$.
>
> We set the other parameters by following our theoretical analysis.
> We set the average weight as $\gamma=1-p$.
> For the momentum parameter, we set $\alpha=1$ for convex-concave case and $\alpha=\max\\{1-\eta\mu/6,1-p\eta\mu/(2\gamma+\eta\mu)\\}$ for strongly-convex-strongly-concave case, where we estimate $\mu$ by $\max\\{\lambda,\beta\\}$ for problem (13). For the sub-problem solver, we set its step-size according to the smoothness parameter of sub-problem (2), i.e., $1/(L+1/\eta)$.
>
> In addition, we estimate the smooth parameter $L$ and the similarity parameter $\delta$ by following the strategy in Appendix C of Ref. [5].
>
> Based on above appropriate setting, our SVOGS achieves better performance than baselines. We are happy to provide the detailed parameters setting in our revision.
>
> **It should be tested on more datasets**
>
> We have provided additional experimental results on datasets w8a ($N=49,749$, $d'=300$) and covtype  ($N=581,012$, $d'=54$). Please refer to the PDF file in Author Rebuttal. We can obverse the proposed SVOGS performs better than baselines on these datasets. We are happy to involve the additional experimental results into our revision.

---

### Official Review · Reviewer_qmRX · 2024-07-13

**Soundness:** 4
**Presentation:** 3
**Contribution:** 4
**Rating:** 7
**Confidence:** 3

**Summary:**

The paper studies distributed min-max optimization under the assumption of second-order data similarity, i.e. the hessians of the objectives at different nodes are close enough. For the classes of (strongly)-convex-(strongly)-concave functions with Lipschitz gradient, lower complexity bounds are proposed. Moreover, an algorithm SVOGS is proposed that strictly reaches the lower bounds in communications and up to a logarithmic factor in local gradient calls. Therefore, the paper almost closes the gap in distributed centralized min-max optimization with data similarity.

**Strengths:**

1. The problem, assumptions and results are clearly stated.
2. An algorithm optimal in communications and near-optimal in gradient calls is proposed. That closes the complexity gap for the class of distributed min-max optimization problems with second-order similarity.
3. Overall, the paper has a readable structure.

**Weaknesses:**

There still remains a gap in the local gradient calls complexity. Maybe it can be overcome with the usage of gradient sliding technique.

**Questions:**

I did not find the difference between the number of communication rounds and communication complexity. It seems that Algorithm 1 sends the same amount of information at each communication round. So why is communication complexity different from the number of communication rounds?

**Limitations:**

The work is theoretical and does not have negative societal impact.

---

> ### Author Rebuttal · Authors · 2024-08-03
>
> **Why is the communication complexity different from the number of communication rounds?**
>
> Recall that the communication complexity in our paper refers to the overall volume of information exchanged among the nodes.
> We take the convex and concave case as an example (Theorem 1) to explain why the communication complexity is different from the number of communication rounds.
>
> 1. The results of Theorem 1 (discussion in Line 147-177) shows we require the communication rounds of $K={\mathcal O}(\delta D^2/\varepsilon)$ to achieve the desired accuracy $\varepsilon$.
>
> 2. We requires the communication complexity of ${\mathcal O}(n)$ to achieve $F(z^0)=\frac{1}{n}\sum_{i=1}^n F_i(z^0)$ in initialization.
>
>
> 3. The index set ${\mathcal S}^k$ with $|{\mathcal S}^k|=b$ means the term of sums in Line 6 of Algorithm 1 takes the communication complexity of ${\mathcal O}(b)={\mathcal O}(\sqrt{n})$ at each iteration.
>
> 4. The communication for the full gradient term $F(w^{k-1})$ in Line 6 of Algorithm 1 does not need to be performed at every iteration.
> Notice that Line 10 performs $w^{k+1}=z^{k+1}$ with probability $p=\Theta(1/\sqrt{n})$ and performs $w^{k+1}=w^{k}$ (no communication) with probability $1-p=1-\Theta(1/\sqrt{n})$,
> which means only the case of $w^{k+1}=z^{k+1}$ needs to communicate to achieve the full gradient with the communication complexity of ${\mathcal O}(n)$.
> Therefore, the overall the expected communication complexity related to the full gradient term is ${\mathcal O}(pn)={\mathcal O}(\sqrt{n})$ at each iteration.
>
> Based on above analysis, we can conclude the overall communication complexity is
> $$ n + {\mathcal O}(b+pn)K = {\mathcal O}(n+\sqrt{n}\delta D^2/\varepsilon),$$
> which is different from the communication rounds
> $$K={\mathcal O}(\delta D^2/\varepsilon).$$
>
>
> In addition, the full participation methods (e.g., EG [24], SMMDS [5] and EGS [25]) requires the communication for the full gradient at every iteration, which leads to the more expensive communication complexity of ${\mathcal O}(n\delta D^2/\varepsilon)$.
>
> **There still remains a gap in the local gradient calls complexity. Maybe it can be overcome with the usage of gradient sliding technique.**
>
> The comparisons in Table 1-2 show the local gradient calls complexities in our results match the lower bounds up to logarithmic factors.
> We thank the reviewer for pointing out that the gap of logarithmic factors could possibly be overcome with the usage of the gradient sliding technique. We believe this is an interesting future direction.

---

> ### Comment · Reviewer_qmRX · 2024-08-09
> **Answer to Rebuttal**
>
> Dear Authors,
>
> Thank you for the response.

---

### Official Review · Reviewer_Vpvu · 2024-07-13

**Soundness:** 3
**Presentation:** 3
**Contribution:** 2
**Rating:** 6
**Confidence:** 4

**Summary:**

This manuscript considers solving (strongly) convex (strongly) concave distributed minimax optimization problem. The authors proposed a stochastic variance-reduced optimistic
gradient sliding method with node sampling, named SVOGS, which achieves complexity nearly matches the obtained lower bound.

**Strengths:**

- The paper improves the complexity of existing algorithms through the adoption of node sampling, especially when dealing with a large number of nodes. The theoretical results are well-supported by simulation experiments.
- The complexity results are thoroughly compared with existing results.
- The paper is well-organized and easy to read.

**Weaknesses:**

- The algorithm design is straightforward. The novelty of the algorithm is limited to the node sampling aspect, which is a simple and obvious adaptation from the existing methods, e.g. Ref. [5]. The authors should further clarify the unique characterisics of the algorithm.
- It seems to the reviewer that incorporating uniform and independent node sampling (c.f., Line 5 in Algorithm 1) does not present any new challenge to the convergence analysis as compared to existing methods. The authors should further clarify the technical contribution of their approach.
- The experiments utilize toy datasets, e.g., a9a, which may limit the generalizability of the results.
- The rationality of Assumption 2 should be discussed in more detail to ensure its validity and relevance to the study.
- Theorems 1 and 2 requires many hyperparameters to be set to exact values that depend on the parameters of the problem. In practice, these parameters are often very difficult to obtain, which may weaken the theoretical results in this work.

**Questions:**

- The authors bring the results for minimization in Ref. [25] directly to compare with the results for the minimax problem in this paper. Does this mean that the minimax problem in this paper does not have additional difficulties compared with the minimization problem?
- The distinction between the optimistic gradient and extra-gradient methods for minimax problem should be clarified. The reviewer is curious about the possibility that the EG method in Ref. [25], combined with node sampling, could also achieve similar results.
- Refer to Weaknesses for more concerns to be addressed properly.

**Limitations:**

The (strongly) convex (strongly) concave distributed minimax optimization problem considered in this work is not applicable to most machine learning tasks and is thus of limited significance in this area.

---

> ### Author Rebuttal · Authors · 2024-08-03
>
> **Results for minimization in Ref. [25]**
>
> Ref. [25] studies both minimization and minimax problems. We only compare their results for the minimax problem (Table 1-3).
>
> The minimax problem is indeed more difficult than the minimization problem. For example, Section 3 of Ref. [25] considers the strongly convex minimization problem by archiving the complexities depend on $\sqrt{\delta/\mu}$ and $\sqrt{L/\mu}$, and Section 4 of Ref. [25] considers the strongly-convex-strongly-concave minimax (strongly monotone variational inequality) problem by archiving the complexities depend on $\delta/\mu$ and $L/\mu$. Noticing that the lower bounds in Section 6.2 of our paper show such dependence on $\delta/\mu$ and $L/\mu$ cannot be improved for this minimax problem, which means it is more difficult.
>
> **Combine EG [25] with node sampling**
>
> Compared with our result, applying the node sampling (variance reduction) [1] to EG method [25] will lead to the additional term $\sqrt{n}$ in the complexity of communication rounds, because EG framework cannot benefit to the mini-batch sampling.
>
> For ease of understanding, we follow the notations and settings of the Algorithm 1 for single machine in Ref. [1] to illustrate this issue (the problem in distributed case is similar). The essential step in the analysis for Algorithm 1 of Ref. [1] is their equation (9) in Lemma 2.9, that is
> $$
> \begin{aligned}
>  & \mathbb{E}_k[2 \tau\langle F\_{\xi_k}(w_k)-F\_{\xi_k}(z\_{k+1 / 2}), z\_{k+1}-z\_{k+1 / 2}\rangle] & \\\\
> \leq & \mathbb{E}_k[2 \tau\\|F\_{\xi_k}(w_k)-F\_{\xi_k}(z\_{k+1 / 2})\\|\\|z\_{k+1}-z\_{k+1 / 2}\\|] & \text { (Cauchy-Schwarz) } \\\\
> \leq & \frac{\tau^2}{\gamma} \mathbb{E}_k[\\|F\_{\xi_k}(z\_{k+1 / 2})-F\_{\xi_k}(w_k)\\|^2]+\gamma \mathbb{E}_k[\\|z\_{k+1}-z\_{k+1 / 2}\\|^2] & \text { (Young's ineq.) } \\\\
> \leq & (1-\alpha) \gamma\\|z\_{k+1 / 2}-w_k\\|^2+\gamma \mathbb{E}_k[\\|z\_{k+1}-z\_{k+1 / 2}\\|^2], & \text { (Assumption 1(iv)) }
> \end{aligned}
> $$
>
> where $\tau=\sqrt{1-\alpha}\gamma/L$ and Assumption 1(iv) is the mean-squared Lipschitz continuous condition on $F_j(\cdot)$ such that $\mathbb{E}\big[\\|F_j(u)-F_j(w)\\|^2\big]\leq L^2\\|u-w\\|^2$ for all $u,w\in\mathcal{Z}$.
> Directly adapting above analysis to our distributed problem will leads to an extra term of $\sqrt{n}$ in the complexity of communication rounds (compared with the methods without node sampling), since the third line in the loop of Algorithm 1 of Ref. [1] only draws one sample $\xi_k$. To match the results of our paper, we desire to introduce the mini-batch sampling like our Algorithm 1 (Line 6) into the framework of EG, which is replacing the term
> $$F_{\xi_k}(w_k)-F_{\xi_k}(z_{k+1/2}) $$
> with
> $$\frac{1}{|\mathcal S^k|}\sum_{j\in\mathcal S^k} (F_{j}(w_k)-F_{j}(z_{k+1/2})),$$
> where $\mathcal S$ follows the notation in our Algorithm 1.
> Then the above derivation becomes
> $$\begin{aligned}
> &\mathbb{E}\_{\mathcal S^k}\left[2 \tau\left\langle\frac{1}{|\mathcal S^k|}\sum_{j\in\mathcal S^k}(F_j(w_k)-F_j(z\_{k+1/2})), z\_{k+1}-z\_{k+1/2}\right\rangle\right]\\\\
> \leq & \mathbb{E}\_{\mathcal S^k}\left[2 \tau\left\\|\frac{1}{|\mathcal S^k|}\sum_{j\in\mathcal{S}^k}(F_j(w_k)-F_j(z\_{k+1/2}))\right\\|\\|z\_{k+1}-z\_{k+1/2}\\|\right]\\\\
> \leq & \frac{\tau^2}{\gamma} \mathbb{E}\_{\mathcal S^k}\left[\left\\|\frac{1}{|\mathcal S^k|}\sum_{j\in\mathcal S^k}(F_j(z\_{k+1/2})-F_j(w_k))\right\\|^2\right]+\gamma \mathbb{E}\_{\mathcal S^k}[\\|z\_{k+1}-z\_{k+1/2}\\|^2]\\\\
> = & \frac{\tau^2}{\gamma} \frac{1}{|\mathcal S^k|^2}\mathbb{E}\_{\mathcal S^k}\left[\sum_{i,j\in\mathcal S^k}\\|F\_{i}(z\_{k+1/2})-F\_{i}(w_k)\\|\\|F_j(z\_{k+1/2})-F_j(w_k)\\|\right]+\gamma \mathbb{E}_k[\\|z\_{k+1}-z\_{k+1/2}\\|^2]\\\\
> = & \frac{\tau^2}{\gamma}\frac{1}{n^2}\sum\_{i,j=1}^n\\|F\_{i}(z\_{k+1/2})-F\_{i}(w_k)\\|\\|F_j(z\_{k+1/2})-F_j(w_k)\\|+\gamma \mathbb{E}_k[\\|z\_{k+1}-z\_{k+1/2}\\|^2]\\\\
> \leq & (1-\alpha)\gamma\\|z\_{k+1/2}-w_k\\|^2+\gamma \mathbb{E}_k[\\|z\_{k+1}-z\_{k+1/2}\\|^2] ,
> \end{aligned}$$
>
> where we use independent uniform  sampling in the second equality and the stronger Lipschitz continuous condition $\\|F_j(u)-F_j(w)\\|\leq L\\|u-w\\|$ for all all $u,w\in\mathcal{Z}$ and $j\in [n]$ (corresponds to Assumption 3 in our paper) in the last inequality.
> Unfortunately, we obverse that the final upper bound cannot be sharpen even if we have introduced the mini-batch sampling like our method, which implies combining the EG method in Ref. [25] and the sampling [1] requires the more iteration numbers (more  communication rounds in distributed case). In contrast, our derivation in the equations on Line 482-484 show the proposed OG-based method can enjoy the benefit from the mini-batch sampling.
>
> **Combing Ref. [5] with node sampling**
>
> Ref. [5] proposed SMMDA, which is based on the FBF framework. Noticing that Ref. [8] proposed TPAPP method by combining node sampling with the FBF framework in Ref. [5]. However, TPAPP cannot achieve the results like our SVOGS.
>
> 1. TPAPP does not consider how to make the duality gap small in the general convex-concave case, which is addressed by SVOGS (see Table 2).
>
> 2. For the measure of distance, the complexities of TPAPP depend on the setting of local iteration number $H$, which leads to its communication complexity and local gradient complexity cannot simultaneously be (near) optimal.
> In contrast, our SVOGS method is simultaneously (near) optimal to all complexities (see Table 2).
>
> 3. In addition, Table 3 shows SVOGS has better complexity to make the gradient small than TPAPP.
>
> **Unique characteristics of the algorithm**
>
> See above discussions.
>
> **Datasets and hyperparameters in experiments**
>
> See Author Rebuttal.
>
> **Rationality of Assumption 2**
>
> The condition of bounded domain is common in the analysis of convex-concave minimax optimization (monotone variational inequality), e.g., Lemma 3 of [1] and equation (24) of [5]. And the practical problem (12) in our experiments indeed satisfies this assumption.

---

> ### Comment · Reviewer_Vpvu · 2024-08-12
>
> Thank you for the detailed response. Since most of my concerns have been well addressed, I raise my overall rating to weak accept.

---

### Author Rebuttal · Authors · 2024-08-03

We thank the reviewers for their appreciation of our work.

Both Reviewer Vpvu and Reviewer DkHm have raised questions about experiments. We provide the response as follows.

**The implementation details (hyperparameters)**

We tune the step-size $\eta$ of SVOGS from $\\{0.01,0.1,1\\}$. The probability $p$ is tuned from $\\{p_0,5p_0,10p_0\\}$, where $p_0=1/\min\\{\sqrt{n}+\delta/\mu\\}$. The batch size $b$ is determined from $\\{\lfloor b_0/10\rfloor,\lfloor b_0/5\rfloor,\lfloor b_0\rfloor\\}$, with $b_0=1/p_0$.

We set the other parameters by following our theoretical analysis.
We set the average weight as $\gamma=1-p$.
For the momentum parameter, we set $\alpha=1$ for convex-concave case and $\alpha=\max\\{1-\eta\mu/6,1-p\eta\mu/(2\gamma+\eta\mu)\\}$ for strongly-convex-strongly-concave case, where we estimate $\mu$ by $\max\\{\lambda,\beta\\}$ for problem (13). For the sub-problem solver, we set its step-size according to the smoothness parameter of sub-problem (2), i.e., $1/(L+1/\eta)$.

In addition, we estimate the smooth parameter $L$ and the similarity parameter $\delta$ by following the strategy in Appendix C of Ref. [5].

Based on above appropriate setting, our SVOGS achieves better performance than baselines. We are happy to provide the detailed parameters setting in our revision.

**More datasets**

We have provided additional experimental results on datasets w8a ($N=49,749$, $d'=300$) and covtype  ($N=581,012$, $d'=54$). Please refer to the PDF file in Author Rebuttal. We can obverse the proposed SVOGS performs better than baselines on these datasets. We are happy to involve the additional experimental results into our revision.

---

### Decision · Program_Chairs · 2024-09-25

**Decision:**

Accept (poster)

**Comment:**

This paper studies convex-concave minimax problems in distributed settings, focusing specifically in settings where the instance satisfies a second order similarity bound. This latter condition refers to an upper bound on the Hessian of the difference between all pairs of objectives within the finite sum. Under such assumption, it is shown that algorithms converge with improved rates, compared to those which simply quantify convergence with smoothness (it should be mentioned that the rates under second-order similarity are never worse than those based on smoothness, from the fact that $L$-smoothness implies $2L$-similarity). The paper also provides matching lower bounds.

Reviewers were mostly positive about this submission. However, it appears that the technical contributions of this work are somewhat marginal:

1. The idea of using gradient sliding appears in [5], and the idea of subsampling nodes is quite standard in distributed optimization.
2. The lower bound is also an adaptation of [42] which is by now quite standard.

Furthermore, the property of Hessian similarity is quite strong. Despite these concerns, and due to the mostly positive assessment of the work, I recommend acceptance.